# ChartSketcher: Reasoning with Multimodal Feedback and Reflection for Chart Understanding

**Muye Huang**[1,2,4], **Lingling Zhang**[1,2],* **Jie Ma**[3], **Han Lai**[1,2], **Fangzhi Xu**[1],
**Yifei Li**[1,2,4], **Wenjun Wu**[1,2], **Yaqiang Wu**[5,1],* **Jun Liu**[1,3]

[1]School of Computer Science and Technology, Xi'an Jiaotong University, Xi'an, China
[2]Shannxi Province Key Laboratory of Big Data Knowledge Engineering, Xi'an, China
[3]MOE KLINNS Lab, Xi'an , China
[4]Zhongguancun Academy, Beijing, China
[5]Lenovo Research
{huangmuye, fangzhixu98, liyifei619584902}@stu.xjtu.edu.cn
{zhanglling, jiema, liukeen}@xjtu.edu.cn
wuyqe@lenovo.com

## Abstract

Charts are high-density visualization carriers for complex data, serving as a crucial medium for information extraction and analysis. Automated chart understanding poses significant challenges to existing multimodal large language models (MLLMs) due to the need for precise and complex visual reasoning. Current step-by-step reasoning models primarily focus on text-based logical reasoning for chart understanding. However, they struggle to refine or correct their reasoning when errors stem from flawed visual understanding, as they lack the ability to leverage multimodal interaction for deeper comprehension. Inspired by human cognitive behavior, we propose ChartSketcher, a multimodal feedback-driven step-by-step reasoning method designed to address these limitations. ChartSketcher is a chart understanding model that employs Sketch-CoT, enabling MLLMs to annotate intermediate reasoning steps directly onto charts using a programmatic sketching library, iteratively feeding these visual annotations back into the reasoning process. This mechanism enables the model to visually ground its reasoning and refine its understanding over multiple steps. We employ a two-stage training strategy: a cold start phase to learn sketch-based reasoning patterns, followed by off-policy reinforcement learning to enhance reflection and generalization. Experiments demonstrate that ChartSketcher achieves promising performance on chart understanding benchmarks and general vision tasks, providing an interactive and interpretable approach to chart comprehension.

## 1 Introduction

Charts are widely used as data visualization methods in scientific papers and business reports. Automated chart understanding is a key step in achieving automated data analysis. Recent advances in MLLMs [1, 24, 38, 45, 50] have shown substantial progress in chart understanding tasks. These include proprietary models like GPT-4o [38] and Gemini-2.0 [44], as well as open-source models such as Qwen-2VL [50] and InternVL-2.5 [4]. The use of MLLMs has become a mainstream approach for chart understanding.

However, existing MLLMs face significant challenges in chart understanding, which involves the systematic interpretation and analysis of visual data representations. Chart understanding requires

---

[1]Corresponding authors. Emails: zhanglling@xjtu.edu.cn; wuyqe@lenovo.com

39th Conference on Neural Information Processing Systems (NeurIPS 2025).

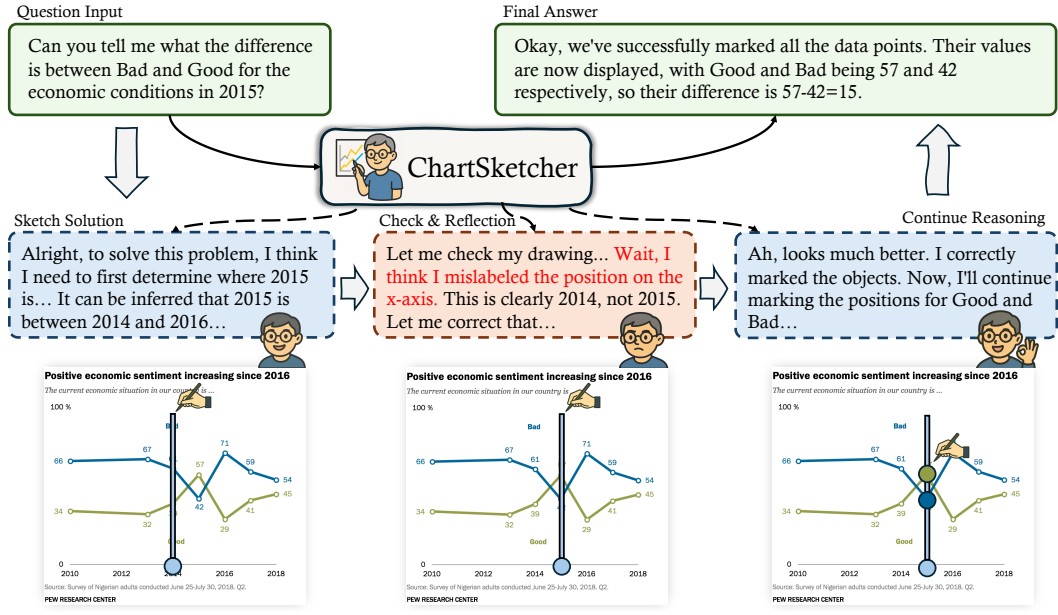

Figure 1: The overview of the proposed ChartSketcher. Dashed lines indicate intermediate reasoning and reflection processes, with corresponding sketch outputs shown for each step.

high-precision visual reasoning capabilities to process complex elements such as overlapping data points, multiple intersecting trend lines, and dense numerical information, demanding simultaneous comprehension of both spatial relationships and their semantic meanings. For example, in Figure 1, answering *"What is the value of 'Good' in 2015?"* requires identifying that 2015 lies between the marked years 2014 and 2016. Models must precisely locate and identify these visual elements while understanding their quantitative relationships to correctly determine that the *'Good'* value in 2015 is 57. This visual reasoning process requires analysis of visual dependencies and precise numerical understanding at each step. Such complex visual reasoning tasks pose significant challenges to existing approaches. MLLMs [3, 5, 47] have attempted to achieve fine-grained visual reasoning through long chains of thought. For example, multimodal reasoning models like QvQ [47] demonstrate the capability to generate long-chain reasoning text. However, their effectiveness remains limited in visually intensive scenarios like charts. This limitation stems from their predominant focus on textual logical processes rather than visual information processing, causing reduced interpretability for users and an inability to correct errors originating from flawed multimodal understanding. Recent attempts, such as VisualCoT [41], Refocus [10] and SketchPad [14], propose image cropping techniques to enhance visual understanding by focusing on key regions. However, the inherent constraints of cropping mechanisms prevent simultaneous analysis of multiple regions, thereby limiting the model's capacity for complex visual reasoning. As illustrated in Appendix A, this represents a challenging case in existing MLLMs. The development of visual reasoning models that focus on processing complex elements requires urgent exploration.

Interestingly, when humans encounter complex visual information, they often create sketches to mark and organize key details. This process helps them break down problems and focus on critical areas within the image. For example, when determining the value of *'Good'*, humans typically start by locating the relevant position on the x-axis, then trace vertically upward to identify the corresponding value on the colored line: a natural way of decomposing the visual reasoning process. This behavior reflects a subconscious strategy humans use to enhance visual focus and understanding.

Drawing inspiration from natural human behaviors, we propose ChartSketcher, a multimodal feedback-driven step-by-step reasoning method that addresses these visual reasoning limitations in chart understanding. Specifically, ChartSketcher employs Sketch-CoT, enabling MLLMs to explicitly annotate their intermediate reasoning processes on images and feed the visual annotations back to themselves, achieving stable step-by-step multimodal reasoning. Moreover, by incorporating

reflection processes between steps and leveraging reinforcement learning, we endow MLLMs with human-like reflection capabilities. The model not only marks the visual reasoning process on images but can also identify reasoning errors and promptly correct mistakes from previous steps. As illustrated in Figure 1, our approach demonstrates powerful visual reasoning capabilities across diverse scenarios. The implementation of ChartSketcher follows a two-stage training pipeline: a cold start phase and an RL phase. In the cold start phase, we transfer reasoning and reflection patterns from LLM to MLLM through cross-modal distillation, creating 300K fine-grained annotated chart understanding samples. The subsequent RL phase employs MCTS and diverse data sampling techniques with over 50K step-by-step reasoning examples to enhance the model's capabilities through off-policy reinforcement learning.

Our main contributions can be summarized in three aspects:

- We propose ChartSketcher, a novel multimodal feedback reasoning approach that enhances MLLMs' visual reasoning capabilities through iterative Sketch-CoT and self-reflection mechanisms. Code and data available at `https://github.com/MuyeHuang/ChartSketcher`.
- We construct a comprehensive dataset of 300K annotated samples for cold start training and 50K curated samples for reinforcement learning. The dataset is designed to support chart step-by-step reasoning.
- We conduct extensive experiments across multiple datasets to demonstrate the effectiveness of ChartSketcher. Through comprehensive ablation studies, we investigate the importance of each training stage and validate the contribution of key components in our work.

## 2    Related Work

**Chart Understanding.** Chart Understanding aims to comprehend the visual context of charts to address specific tasks, such as QA or summarization. FigureQA [18] stands as a pioneering work, introducing a chart understanding pipeline capable of handling binary classification tasks for chart-related questions. Subsequent works [29, 22, 26, 23] further enhanced chart understanding capabilities by employing multi-component pipelines. For instance, DePlot [63] leveraged multiple components combined with the mathematical capabilities of LLMs to achieve performance improvements on PlotQA [36]. With the advent of MLLMs, approaches utilizing MLLMs as the primary component have become mainstream in the field [31, 32, 2, 15, 53, 52]. ChartLlama [12], through clever data construction and fine-tuning of LLaVA [24], built a robust chart-expert model. Leveraging the powerful language capabilities inherent in MLLMs, recent studies have employed multi-task training methodologies to bolster chart understanding across a variety of tasks. ChartAssistant [35] utilized unified multi-task training to improve overall performance. TinyChart [62] utilizes the Program-of-Thoughts technique to enhance numerical reasoning capabilities in chart QA tasks. ChartMoE [57] employed a Mixture of Experts approach to model different chart types effectively, thereby enabling understanding across diverse chart categories.

**Multimodal Reasoning.** Models such as OpenAI-o1 [37] and Deepseek-R1 [7] have demonstrated the strong reasoning capabilities of LLMs [11, 58, 40, 54, 48, 55, 28], often enhanced through RL. However, the reasoning capabilities of MLLMs remain an area of active investigation. Current approaches often focus on training MLLMs using CoT techniques [51, 59] to generate step-by-step reasoning sequences across diverse tasks. These methods [8, 25, 27, 6] predominantly concentrate on CoT techniques within the textual modality, relying heavily on the MLLM's underlying LLM backbone to perform multi-step inference. VisualCoT [41] introduced a method involving cropping critical regions to aid the model in focusing on pertinent visual areas. While prior work predominantly focuses on text-based CoT methods or region-limited approaches, these techniques struggle with scenarios requiring attention to multiple distinct visual elements. Our work addresses this limitation by integrating self-prompted visual markers and multimodal feedback into the CoT process, enabling more comprehensive and robust multimodal reasoning.

## 3    ChartSketcher

We propose ChartSketcher, which enables step-by-step reasoning in multimodal chart understanding by sketching directly on chart images. In the following sections, we will present the implementation details of ChartSketcher, including its architecture and training specifics.

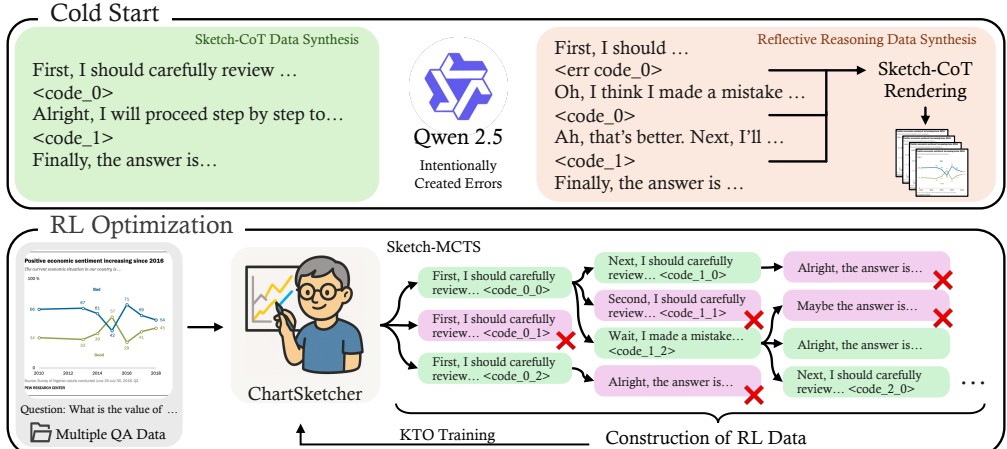

Figure 2: Overview of ChartSketcher Training Process. The upper part illustrates the cold start phase, focusing on knowledge distillation and pattern learning. The lower part shows the offline RL optimization process, which is conducted on diverse datasets. In the figure,  indicates that ChartSketcher is calling the Programmatic Sketching Library to draw. When ChartSketcher no longer outputs , it indicates that the reasoning process has ended.

## 3.1 Architecture

Enabling the model to reason while sketching, much like a human, is the core objective of ChartSketcher. To enable the MLLM to perform Sketch-CoT reasoning, we designed two integrated modules: a programmatic sketching library and a sketching reasoning pipeline. The details of these two modules are introduced as follows:

**1) Programmatic Sketching Library.** To equip MLLMs with image sketching capabilities, we design a simple drawing language library. The library provides basic operations to create and manipulate geometric shapes (points, lines, circles, arrows, and their combinations) through simple command syntax. During the reasoning process, the MLLM can insert drawing commands at any position to create new visual elements or modify existing ones through operations like translation, rotation, and deletion. The detailed command specifications, library guide, and supported operations are listed in Appendix B.

**2) Sketching Reasoning Pipeline.** In the process of Sketch-CoT reasoning, it is necessary to view the draft content in real time to ensure the continuity of reasoning. We have designed a visual feedback pipeline that parses the output of the MLLM, generates sketches, and feeds the newly rendered image back to the MLLM. This new image is then fully re-encoded to generate new visual tokens for the next reasoning step, forming a closed loop of action, perception, and reflection. This process operates as a "reflection and draw-feedback" loop, as illustrated in Figure 1, which terminates upon the completion of reasoning, where no further sketching code is produced, and the pipeline exits the loop automatically.

## 3.2 Training Process

ChartSketcher implements Sketch-CoT reasoning through the multi-turn dialogue mechanism of MLLMs. Therefore, ChartSketcher primarily focuses on understanding the differences between sequential images in a dialogue. Existing MLLMs lack capabilities for such serialized visual understanding; therefore, we introduce a two-stage process comprising cold and RL optimization. Cold start focuses on learning the reasoning patterns for multi-turn visual feedback, RL optimization leverages an off-policy RL approach to further enhance reasoning capabilities. The following sections will detail the construction of training data and the specifics of the training process for these steps.

### 3.2.1 Cold Start

Cold start is designed to learn the Sketch-CoT reasoning patterns with visual feedback. It is divided into two steps: the first step trains the model to understand and generate sequential visual reasoning patterns, enabling it to process multi-turn visual information coherently, while the second step builds upon this foundation by developing reflective reasoning capabilities that allow the model to evaluate and improve its solutions based on visual feedback.

**Sketch-CoT Data Synthesis.** This step aims to synthesize rich Sketch-CoT data, which is used to train the model's reasoning patterns and its ability to read visual feedback. The detailed process is illustrated in Appendix C. Specifically, the construction process is divided into the following three steps:

**1) Question Construction:** We used the EvoChart-Corpus [15], a dataset with high-quality synthesized chart images. While it provides chart images, its QA pairs are template-generated, which may limit the diversity of reasoning chains. Therefore, we used all the images and part of the QA pairs from EvoChart-Corpus. We then developed a seed-based method to create more diverse questions. We used QA pairs from existing ChartQA datasets as seeds to prompt the LLM to generate similar new questions based on the EvoChart-Corpus image and annotations. This approach helped us create many diverse and meaningful questions.

**2) Reasoning Process Construction:** With annotations available, multimodal questions can be converted into simpler text-based questions. Similarly, visual reasoning chains can also be converted into textual reasoning chains. Inspired by this, we distilled the reasoning capabilities of the LLM into the MLLM. Specifically, we input the questions and detailed annotations of the EvoChart-Corpus into the LLM and prompted the model to output reasoning chains. We enforced a rule that the LLM must output sketching code to justify its conclusions before providing any final or intermediate conclusions. This ensures that the constructed multimodal reasoning chains are factually grounded.

**3) Rendering:** We used the sketching code in the reasoning chains as a boundary to segment the reasoning process into multiple steps, adding visual feedback between these steps. All prompts can be found in the Appendix C. Using the above methods, we constructed over 300k Sketch-CoT samples.

**Reflective Reasoning Data Synthesis.** This step aims to synthesize Sketch-CoT data with reflective reasoning processes, training the model to identify errors from visual feedback and correct them in a timely manner. To enable reflection, we manually construct erroneous reasoning processes. For simplicity, the reflective reasoning data is built based on the previously generated correct Sketch-CoT data. Specifically, the process involves the following steps:

**1) Reflective Construction:** We prompt the LLM to introduce an error in a specific step by providing incorrect drawing coordinates. The error types can vary, such as using coordinates from other points, coordinate drift, and more. Subsequently, the LLM reflects on and corrects these coordinates, providing new drawing code. During the reflection process, the LLM uses conversational expressions and human-like interjections, such as "Oh" or "Hmm," to mimic natural reasoning. Detailed examples can be found in Figure 2.

**2) Data Mixing:** Through the above steps, each Sketch-CoT generates two versions: one with reflection and one without. These two versions share the same CoT prefix. At the final step of the prefix, if erroneous data is selected, it constitutes reflective data; otherwise, it is non-reflective data. We mix reflective and non-reflective data at a 1:1 ratio, encouraging the model to reflect only when errors occur, with greater focus on visual feedback information.

### 3.2.2 RL Optimization

After undergoing a cold start phase, ChartSketcher learns patterns from annotated synthetic data but has not been trained on real-world unlabeled datasets. To enhance the generalization ability of ChartSketcher, we incorporate off-policy RL optimization inspired by works on natural language. We designed an off-policy RL strategy based on a variant of MCTS, sketch-MCTS, which can collect high-quality RL data on datasets without bbox annotations. Our approach identifies optimal paths by evaluating the average $Q/N$ value of each potential solution path, selecting nodes along the optimal trajectory as positive samples while designating low-value siblings, nodes with rendering errors, and duplicate nodes as negative samples. To maintain sample quality, we exclude siblings with

positive values above zero from the negative sample pool. This strategic sampling mechanism enables effective learning from unlabeled data while preserving the model's discriminative capabilities. The following sections detail the sketch-MCTS algorithm that underpins this RL optimization framework.

**Sketch-MCTS Algorithm.** MCTS is a multi-step reasoning algorithm with single-step action output, which implicitly considers the consequences of multi-step decision making. Our proposed sketch-MCTS collects all implicit processes and identifies the potentially optimal answers. The formal representation is shown in Appendix D. Sketch-MCTS modifies the original MCTS algorithm while retaining its core principles, enabling the generation of a complete search tree in a single run. Specifically, we made the following modifications to the MCTS algorithm:

**1) Modifications to the Expansion step:** We set rules to control the behavior of the Expansion step in order to obtain more diverse paths. Expansion generates as many potential next steps as possible by using high temperature. To limit redundant paths, we set a deduplication mechanism: if two nodes generate identical drawing codes, the duplicate node is directly removed. Additionally, to prevent invalid reflection, if the drawing code of a child node is a subset of its parent node's drawing code, the child node is considered an invalid reflection node, as it results in ineffective changes.

**2) Handling of leaf nodes:** In our method, the criteria for determining leaf nodes are more stringent and explicit. A node is only marked as a leaf node if its drawing code contains errors or if it fails to generate any drawing code (indicating the end of the solution). Furthermore, once a node is identified as a leaf node, its correctness is immediately evaluated, and the reward is backpropagated.

**3) Conditions for terminating the search:** To prevent infinite searches for complex problems and excessive searches for simple problems, we designed dual termination conditions: 1) The search ends when a certain number of correct answers are found in the search tree. 2) The search also terminates when the number of simulations exceeds a predefined threshold. This dual condition adapts to problems of varying difficulty: for simple problems, the algorithm converges quickly, while for complex problems, the algorithm can perform sufficient exploration within a reasonable range.

## 4 Experiments

### 4.1 Settings

**Data Construction.** During the cold start phase, our base dataset images were sourced from EvoChart-Corpus, with seed questions from ChartQA [30] and EvoChart-QA [15]. To ensure general capability, we incorporated 20% of VisualCoT [41] data into the training mix. For the RL phase, we conducted training across multiple datasets, including ChartQA, ChartBench [56], and VisualCoT. For model selection, we employed Qwen2.5-32B [40] to construct QA pairs and distill multimodal reasoning and reflection data. We used DeepSeek-Distill-Qwen-14B [7] as the value network for Sketch-MCTS, evaluating the correctness of final answers. We also trained a smaller, 2B version ChartSketcher-2B to facilitate its use in scenarios with limited computational resources. ChartSketcher-72B and ChartSketcher-2B were initialized with Qwen2VL-72B [50] and Qwen2VL-2B weights, respectively. All prompts used in data construction are detailed in the Appendix C. We tested chart understanding capabilities on ChartQA and other datasets [36], and evaluated general performance on Openimages and other datasets [43, 34, 17, 20, 61, 64, 19]. The detailed composition of our training data is presented in Table 1.

Table 1: Detailed breakdown of the training data used in each phase of ChartSketcher's development.

| Phase | Method | Data Source | Data Type | Quantity |
|---|---|---|---|---|
| Cold Start | SFT | EvoChart Synthetic Chart Data
VisualCoT and its Annotations | Correct Reasoning Path
Correct Reasoning Path | 155,203 (87.3%)
22,510 (12.7%) |
| | | | *SFT Subtotal* | *177,713* |
| | DPO | EvoChart Synthetic Chart Data | Reflection Reasoning Path | 147,955 |
| RL | KTO | ChartQA and ChartBench
General QA-Pairs* | MCTS Sampled Paths
MCTS Sampled Paths | 41,196 (81.6%)
9,259 (18.4%) |
| | | | *RL Subtotal* | *50,455* |

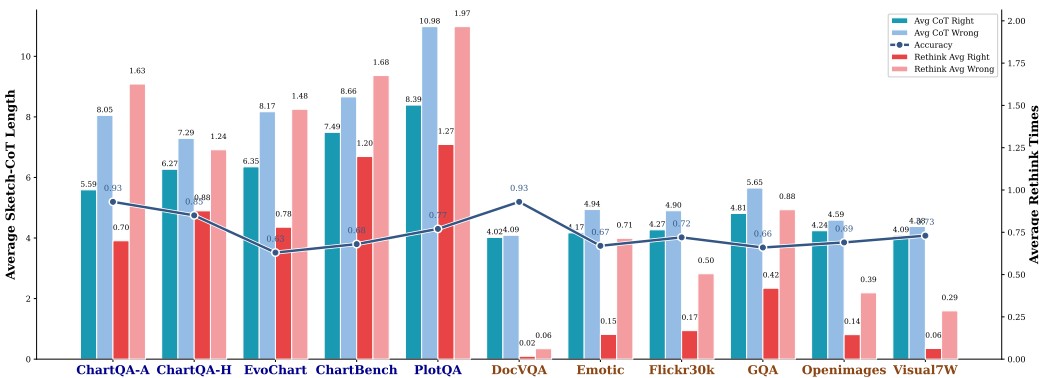

Figure 3: Analysis of CoT length and the number of rethink iterations for both correctly and incorrectly answered questions across all listed datasets. Datasets listed on the left (blue font) are chart-specific benchmarks, while those on the right (brown font) represent general image datasets.

**Evaluation Metrics.** To accurately evaluate model performance, we employ DeepSeek-Distill-Qwen-32B [7] to assess the alignment between MLLM outputs and the QA dataset answers. To mitigate model variance, we adopt a voting mechanism where each question is evaluated 3 times, and a correct answer is determined by a majority vote threshold of 2. To ensure fair comparison, all experimental results reported in this paper are based on our local reproduction of baseline methods.

**Training Settings.** During the cold start phase, we trained ChartSketcher for 4 epochs on data without reflection, followed by 1 epoch using RPO [60] loss on reflection data. To reduce computational costs, we employed LoRA [13] training in the cold phase, with a LoRA rank of 16, Alpha of 32, batch size of 64, and learning rate of 1e-4. The RPO ratio was set to 1.0. In the RL phase, we conducted KTO [9] training for 1 epoch, maintaining a LoRA rank of 16 and Alpha of 32, while adjusting the batch size to 32 and reducing the learning rate to 1e-5. For the key parameters of MCTS, the maximum tree depth is 8, the maximum number of child nodes is 3, $C_{PUCT} = 3.0$, the simulation count limit is 15, and the search exits after successfully finding 3 answers. All experiments were run on two machines: an Atlas 800T A2 and 8 * A800-40G GPUs. For more training details, see supplementary materials.

## 4.2 Performance Comparison

Table 2 presents the complete results for the chart-specific benchmarks alongside selected results from the general datasets. Compared to the baseline Qwen2VL-72B, our proposed ChartSketcher-72B exhibits significant improvements across both chart-specific and general-purpose datasets. Notably, even when compared to QvQ-Preview and GPT-4o, our method still maintains an advantage on the chart-specific datasets. Meanwhile, Figure 4 demonstrates that our method offers richer interactivity. Furthermore, ChartSketcher-2B achieves substantial improvements over its baseline Qwen2VL-2B and chart expert model ChartGemma [32] across nearly all evaluated datasets. This demonstrates that our approach effectively enhances performance in the specialized chart domain without significantly compromising its general-purpose capabilities. Moreover, as shown in Figure 4, our method demonstrates better user-friendliness and greater interpretability compared to other approaches. The complete evaluation results on all 18 datasets can be found in Appendix E.

## 4.3 Ablation Study

Our ablation study is conducted based on the ChartSketcher-72B model. Specifically, we investigate the following configurations:

- *w/o Rethink & RL*: This setting omits the Rethink learning step during the cold start phase. The model proceeds directly with MCTS sampling followed by SFT.

- *w/ Rethink w/o RL*: This setting includes the complete cold start phase (with Rethink learning), but replaces the subsequent RL phase with SFT.

Table 2: Experimental results on chart and vision benchmarks. PlotQA reports sampled results, and VisualCoT shows a composite score across multiple datasets.

| Model | ChartQA-H | ChartQA-A | EvoChart-QA | ChartBench | PlotQA | VisualCoT |
|---|---|---|---|---|---|---|
| *Proprietary models* | | | | | | |
| GPT-4o | 84.32 | 88.48 | 52.80 | **61.47** | 42.96 | **78.45** |
| Gemini-2.0 | 84.00 | 88.24 | **64.64** | 55.63 | **63.36** | 77.90 |
| Claude-3.5 | **85.04** | **90.72** | 56.96 | 56.96 | 57.63 | 75.93 |
| *Open-source models and chart expert models* | | | | | | |
| Qwen2VL-72B | 82.48 | 88.56 | 54.00 | 54.77 | 73.76 | 72.14 |
| QvQ-Preview-72B | 83.20 | 89.76 | 54.32 | 42.40 | 69.04 | 76.52 |
| InternVL2.5-78B | 78.48 | 89.44 | 57.44 | 65.57 | 57.20 | **78.93** |
| ChartGemma-2B | 53.44 | 86.64 | 36.08 | 23.87 | 25.76 | 55.62 |
| Qwen2VL-2B | 50.48 | 75.84 | 23.84 | 20.27 | 38.80 | 58.33 |
| ChartSketcher-2B | 55.60 | 80.88 | 26.72 | 30.10 | 41.12 | 66.86 |
| ChartSketcher-72B | **85.20** | **92.64** | **63.28** | **68.33** | 76.72 | 76.59 |
| *Ablation study based on ChartSketcher-72B* | | | | | | |
| w/o Rethink & RL | 77.76 | **91.12** | 51.12 | 50.40 | 67.76 | 72.58 |
| w/ Rethink w/o RL | 76.64 | 90.56 | 51.36 | 52.93 | 67.68 | 70.95 |
| w/o RL | 77.12 | 88.80 | 39.84 | 52.73 | 67.84 | 68.89 |
| w/o Feedback | **81.52** | 91.04 | **57.76** | **56.13** | **72.24** | 75.18 |
| w/o CoT | 75.12 | 90.08 | 55.36 | 47.43 | 68.16 | **76.12** |

| Model | Openimages | Flickr30k | DocVQA | Visual7W | GQA | Emotic |
|---|---|---|---|---|---|---|
| *Proprietary models* | | | | | | |
| GPT-4o | 52.49 | 79.04 | 94.93 | **77.60** | 68.30 | **53.81** |
| Gemini-2.0 | 57.78 | **79.34** | 95.27 | 77.20 | **68.51** | 41.22 |
| Claude-3.5 | **62.50** | 75.68 | **97.64** | 73.70 | 60.63 | 35.42 |
| *Open-source models and chart expert models* | | | | | | |
| Qwen2VL-72B | 51.75 | 61.64 | 93.36 | 73.90 | 57.06 | 43.14 |
| QvQ-Preview-72B | 60.21 | 72.96 | 92.68 | 69.90 | 63.91 | 65.55 |
| InternVL2.5-78B | 60.85 | **76.39** | 95.16 | 74.40 | **72.19** | 61.59 |
| ChartGemma-2B | 49.21 | 57.37 | 57.32 | 57.30 | 51.33 | 53.20 |
| Qwen2VL-2B | 53.97 | 34.48 | 79.50 | 60.40 | 23.31 | 50.15 |
| ChartSketcher-2B | 64.23 | 68.95 | 69.82 | 64.90 | 61.25 | 59.45 |
| ChartSketcher-72B | **68.68** | 72.19 | 92.68 | 73.00 | 65.85 | **67.16** |
| *Ablation study based on ChartSketcher-72B* | | | | | | |
| w/o Rethink & RL | 62.33 | 66.04 | 89.08 | 65.80 | 62.17 | 58.54 |
| w/ Rethink w/o RL | 59.05 | 66.17 | 88.40 | 66.00 | 61.96 | 58.38 |
| w/o RL | 57.57 | 66.43 | 86.94 | 62.50 | 57.87 | 53.51 |
| w/o Feedback | **67.94** | **70.96** | **92.57** | **70.80** | **65.54** | 54.88 |
| w/o CoT | 62.43 | 69.53 | 92.23 | 67.20 | 62.88 | **64.70** |

- *w/o RL*: This represents the ChartSketcher model after completing only the cold start phase, without undergoing the RL phase.

- *w/o Feedback*: In this setting, the multimodal image feedback mechanism is disabled during inference. An empty string is used as a placeholder for the multimodal feedback input.

- *w/o CoT*: This baseline setting does not apply the ChartSketcher methodology. Instead, the model is fine-tuned using SFT on the identical dataset used for training ChartSketcher.

Based on the results presented in Table 2, we can draw the following conclusions:

**1) Sketch-CoT is effective.** Observing the *w/o CoT* setting reveals a significant performance decline compared to the baseline when Sketch-CoT is not employed. This highlights the crucial role of our CoT approach guided by sketches and multimodal feedback.

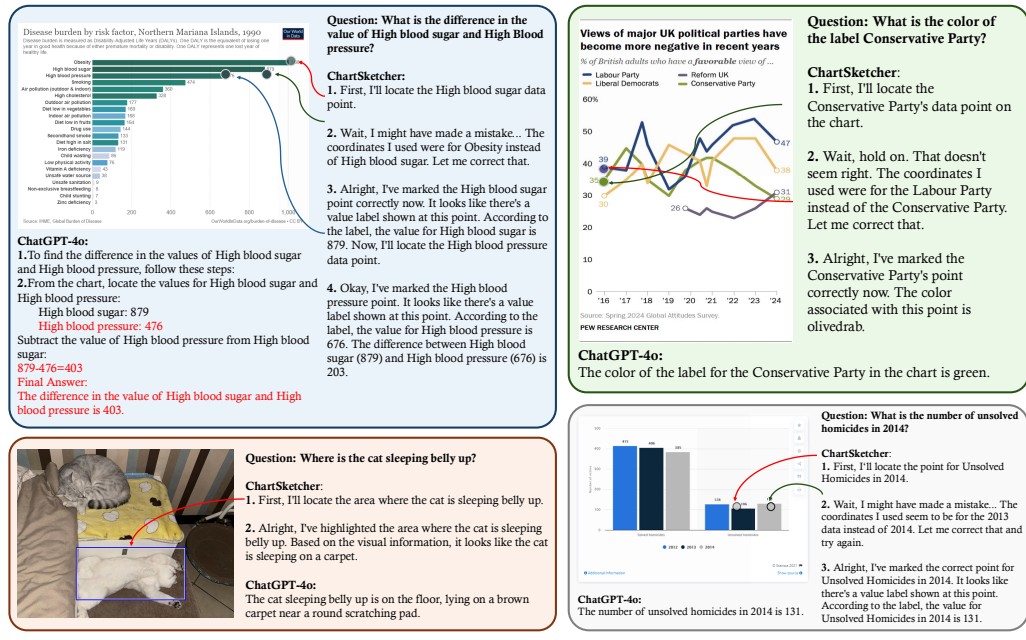

Figure 4: Four cases for ChartSketcher. The drawing code associated with each step is omitted for clarity. Arrows indicate the visual outputs generated by specific reasoning steps. Semi-transparent elements represent outputs that were subsequently corrected or erased by later steps.

**2) The Rethink and RL phases are critical components of ChartSketcher.** Removing the Rethink step or replacing RL with SFT, as seen in the *w/ Rethink w/o RL* setting, results in a comprehensive drop in performance across the board. Notably, substituting RL with SFT leads to a substantial decrease in general-purpose capabilities, evidenced by the VisualCoT aggregate score dropping to 70.95%, which is below the baseline performance.

**3) The two-stage ColdStart-RL training methodology effectively enhances model capabilities.** For instance, the model after only the cold start phase (*w/o RL*) performs slightly below the baseline. This is expected, as the cold start phase primarily utilizes Out-Of-Distribution (OOD) synthetic data. However, the subsequent RL phase rapidly elevates the model's performance, surpassing not only the baseline but even the QvQ-Preview model.

**4) Multimodal feedback plays a significant role in multimodal reasoning.** As indicated by the *w/o Feedback* setting, the absence of feedback has a relatively minor impact on datasets that emphasize direct information extraction with less need for complex multimodal reasoning, such as ChartQA-A and the overall VisualCoT score. However, for datasets demanding intricate multimodal reasoning, like EvoChart and ChartQA-H, removing feedback leads to a more pronounced performance degradation.

### 4.4 Analysis of Reasoning Steps

To quantitatively analyze the general patterns regarding the length of reasoning and the frequency of rethinking employed by ChartSketcher when addressing different questions, we examined the average CoT length and the average number of rethink iterations for both correctly and incorrectly answered questions within the evaluation sets. The results are depicted in Figure 3. We derive the following findings:

**1) More challenging questions elicit longer CoTs.** As observed in Figure 3, the average CoT length for the more demanding datasets, EvoChart-QA and ChartBench, reaches 6-7 steps; both require complex chart reasoning. The CoT length for correctly answered questions is consistently shorter than that for incorrectly answered questions across all datasets. This indicates that the model tends to employ more reasoning steps for difficult problems and attempts to refine answers through multiple rethink iterations when facing challenges.

**2) The model utilizes rethinking to identify potential errors.** Across all datasets shown in Figure 3, the average number of rethink iterations is higher for incorrect answers than for correct ones. This suggests that when faced with difficult problems, the model not only extends the CoT through multi-step reasoning but also actively employs the rethink mechanism in an attempt to revise its solution. This occurs even if the model ultimately fails to provide the correct answer, demonstrating persistent attempts at self-correction.

**3) Chart-specific datasets demand more complex reasoning processes compared to general-purpose datasets.** Within the chart-specific benchmarks, even ChartQA-H, which has the highest accuracy among them (implying relative simplicity), exhibits an average CoT length greater than that of GQA, the general-purpose dataset with the longest average CoT. This demonstrates ChartSketcher's capability to engage in complex, multi-step CoT reasoning, potentially involving multiple rethink iterations, to achieve precise inference specifically within the demanding chart domain.

### 4.5 Case Visualization

We select four representative examples to illustrate the visual reasoning capabilities of ChartSketcher, as depicted in Figure 4. These cases demonstrate ChartSketcher's ability to identify errors within its own reasoning steps and implement timely corrections. The example presented in the top-right indicates that ChartSketcher can rectify single-step mistakes while concurrently executing multi-step numerical extraction and computation tasks. Interestingly, the bottom-right example reveals that despite being a model specialized for charts, ChartSketcher retains a significant capacity for understanding natural images. This finding broadens the potential application scope and highlights the versatility of ChartSketcher. For more detailed case visualizations, please refer to Appendix F.

## 5 Conclusion

We presented ChartSketcher, a novel multimodal feedback-driven approach for chart understanding. By enabling MLLMs to visually sketch charts during their reasoning process through a programmatic sketching library, our method more closely mirrors human cognitive behavior in visual analysis tasks. The two-stage training strategy combines cross-modal distillation and reinforcement learning with Sketch-MCTS, which allows the model to effectively learn and refine sketch-based reasoning chains. Experimental results demonstrate that ChartSketcher's integration of visual feedback and iterative refinement outperforms existing methods on various chart understanding benchmarks. While our empirical results are strong, a formal theoretical analysis of the convergence and robustness of the sketch-feedback loop remains an important avenue for future research. Future work could also explore expanding the sketching capabilities and feedback mechanisms to tackle even more complex visual reasoning scenarios, such as 3D or dynamic charts.

## Acknowledgments and Disclosure of Funding

This work was supported by the National Key Research and Development Program of China (2022YFC3303600), National Natural Science Foundation of China (No. 62137002, 62293550, 62293553, 62293554, 62450005, 62437002, 62477036, 62477037, 62176209, 62192781, 62306229), 'LENOVO-XJTU' Intelligent Industry Joint Laboratory Project, the Shaanxi Provincial Social Science Foundation Project (No. 2024P041), the Natural Science Basic Research Program of Shaanxi (No. 2023-JC-YB-593), the Youth Innovation Team of Shaanxi Universities 'Multi-modal Data Mining and Fusion', Project of China Knowledge Center for Engineering Science and Technology, the Zhongguancun Academy Project (No. 20240103), the Youth AI Talents Fund of China Association of Automation (Grant No._HBRC-JKYZD-2024-311).

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

## Appendix Contents

## A  Challenging Case in Current MLLMs

We conducted comprehensive case testing on multiple multimodal language models, including QvQ-Preview, OpenAI-o3-mini, GPT-4o, and Kimi-1.5-pro [46]. As illustrated in Figure 5, our analysis revealed that despite possessing text reasoning capabilities, both QvQ and OpenAI-o3-mini produced erroneous responses even after extended deliberative processes. Given that o3's reasoning mechanisms are not externally observable, we focused our analytical efforts on QvQ's reasoning trajectory. This examination demonstrated that despite generating elaborate chains of reasoning, QvQ consistently failed to overcome fundamental visual perception limitations—specifically, the inability to verify the presence of "2015" within the image. Additionally, our findings indicate that the performance of more advanced models such as GPT-4o and Kimi-1.5-pro remains sub-optimal in this context, as they demonstrate significant deficiencies in comprehending critical visual information essential for accurate task completion.

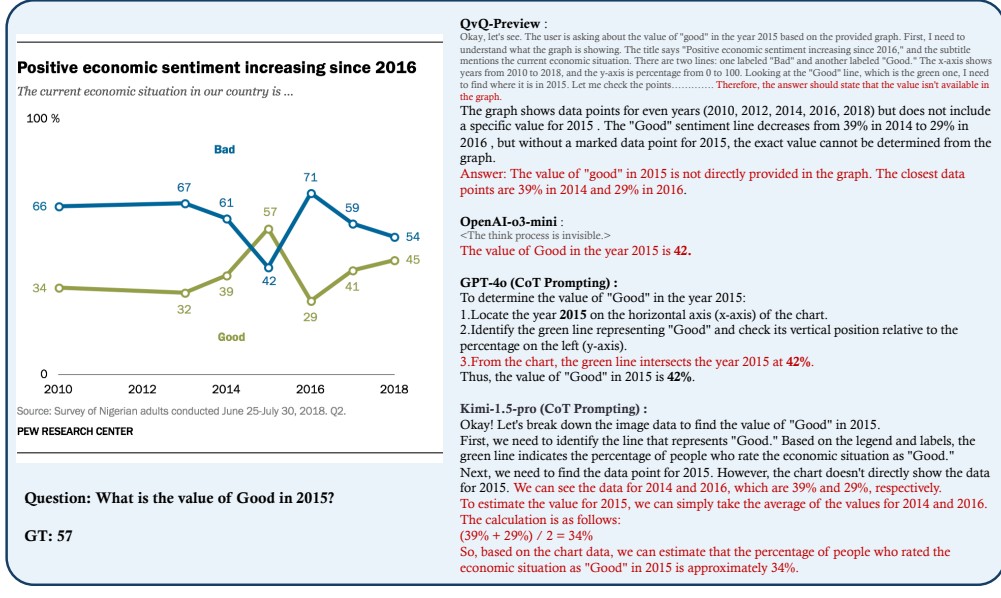

Figure 5: An illustrative example of a challenge in current MLLMs

## B  Programmatic Sketching Library

To equip MLLMs with advanced image sketching capabilities, we designed a lightweight and versatile drawing language library. This library supports operations to create and manipulate basic geometric shapes—such as **points, lines, circles, arrows**, and their combinations—through a simple and intuitive command syntax. During the reasoning process, MLLMs can dynamically insert drawing commands to generate new visual elements or modify existing ones using operations like **translation, rotation, and deletion**. Below, we detail the pseudocode structure, supported commands, and their usage.

## B.1  Pseudocode Overview

The pseudocode serves as a structured and concise language for defining geometric shapes and applying transformations. It operates within a normalized coordinate system: - The top-left corner of the canvas corresponds to $(0,0)$, and the bottom-right corner corresponds to $(1,1)$. - The horizontal axis is denoted by $x$, and the vertical axis by $y$.

The pseudocode executes commands line-by-line, starting with the `BEGIN` keyword and terminating at `END`. Any commands written after `END` are ignored.

---

**Key Features of the Pseudocode**

- **Execution Blocks:** Commands are executed between `BEGIN` and `END`. Lines outside this block are ignored.
- **Normalized Coordinates:** The canvas is scaled to a unit square with $(0,0)$ at the top-left and $(1,1)$ at the bottom-right.
- **Dynamic Operations:** Shapes can be created, modified, and deleted in real-time through simple commands.
- **Geometric Flexibility:** Supports points, lines, circles, rectangles, and arrows, covering a wide range of visual elements.

---

## B.2  Supported Commands

### B.2.1  Shape Creation

The library allows for the creation of several geometric shapes. Below are the commands and their specific syntaxes.

---

**Shape Creation Commands**

- **Point:** `create_point entity_id x y color` Creates a point at coordinate $(x, y)$ with the specified `color`. **Example:** `create_point p1 0.2 0.2 red`
- **Line:** `create_line entity_id x1 y1 x2 y2 color` Creates a line connecting $(x_1, y_1)$ and $(x_2, y_2)$ with the specified `color`. **Example:** `create_line l1 0.2 0.2 0.8 0.8 blue`
- **Circle:** `create_circle entity_id cx cy radius color` Creates a circle centered at $(cx, cy)$ with radius `radius` and the specified `color`. **Example:** `create_circle c1 0.5 0.5 0.1 green`
- **Rectangle:** `create_rectangle entity_id x1 y1 x2 y2 color` Creates a rectangle with the top-left corner at $(x_1, y_1)$ and bottom-right corner at $(x_2, y_2)$. **Example:** `create_rectangle r1 0.1 0.1 0.4 0.4 black`
- **Arrow:** `create_arrow entity_id x1 y1 x2 y2 color` Creates an arrow from $(x_1, y_1)$ (tail) to $(x_2, y_2)$ (head). **Example:** `create_arrow a1 0.3 0.3 0.7 0.7 purple`

---

### B.2.2  Transformation Operations

The library supports the following operations to manipulate existing shapes:

---

**Transformation Commands**

- **Translation:** `translate entity_id dx dy` Moves the shape identified by `entity_id` by $(dx, dy)$. **Example:** `translate l1 0.1 0.1`
- **Rotation:** `rotate entity_id angle cx cy` Rotates the shape identified by `entity_id` around the point $(cx, cy)$ by `angle` degrees. **Example:** `rotate l1 45 0.5 0.5`
- **Deletion:** `delete entity_id` Deletes the shape identified by `entity_id`. **Example:** `delete l1`

---

### B.2.3 Program Control

Special commands control the execution of the pseudocode:

- **Begin Command:** `BEGIN` Marks the start of pseudocode execution. All commands following this are executed until `END` is encountered. **Example:** `BEGIN`

- **End Command:** `END` Terminates pseudocode execution. Commands after `END` are ignored. **Example:** `END`

### B.3 Example Pseudocode

The following example illustrates the creation and transformation of shapes using the pseudocode:

```
Example Pseudocode

BEGIN
create_point p1 0.2 0.2 red
create_line l1 0.2 0.2 0.8 0.8 blue
create_circle c1 0.5 0.5 0.1 green
create_arrow a1 0.3 0.3 0.7 0.7 purple
translate l1 0.1 0.1
rotate l1 45 0.5 0.5
END
create_rectangle r1 0.1 0.1 0.4 0.4 black
```

**Explanation:** The above pseudocode performs the following steps:

- Creates a `red` point, a `blue` line, a `green` circle, and a `purple` arrow.

- Translates the line by $(0.1, 0.1)$ and rotates it $45°$ around the center $(0.5, 0.5)$.

- The rectangle defined after `END` is ignored.

### B.4 Frequently Asked Questions

1. **How can I change the color of a shape?** Specify the color in the creation command. For example: `create_point p1 0.2 0.2 red`

2. **What does** `END` **do?** The `END` command stops the pseudocode execution. Commands after `END` are ignored.

3. **What happens if I forget** `END`**?** If `END` is missing, the parser continues parsing until the last line. Always include `BEGIN` and `END`.

4. **How do translation and rotation work?** - **Translation:** Moves the shape by $(dx, dy)$. - **Rotation:** Rotates the shape around a specified center $(cx, cy)$ by a given angle.

5. **How do I delete a shape?** Use the `delete` command with the shape's identifier. For example: `delete l1`

## C  Detailed Process of Sketch-CoT Data Synthesis

As shown in Figure 6, we present the data synthesis process during the cold start stage. The process begins with the synthesis of data without reflection, as the synthesis of reflection-based data relies on the initial non-reflective data.

In the non-reflective data synthesis process, we use seed techniques to augment QA pairs. Subsequently, the reasoning process is distilled using Qwen2.5-32B. The distillation prompt is formally defined in **Prompt C** below.

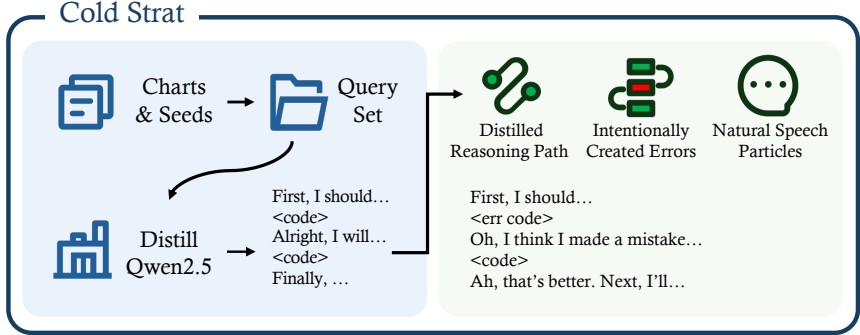

Figure 6: The synthesis process of Sketch-CoT. The left side illustrates the CoT synthesis process without reflection, while the right side demonstrates the synthesis process of CoT with reflection.

---

**Prompt for Distilling Non-Reflective Sketch-CoT**

Your task is to output a simulated human-like reasoning dialogue. Since you cannot draw directly, all drawing operations must be expressed in pseudocode enclosed between the keywords `BEGIN` and `END`. The following are the requirements for the simulated dialogue:
- Do not reveal the final answer before completing the reasoning process. The answer must be derived from the reasoning steps. - The `BEGIN` and `END` markers should be embedded within the text, and I will parse them automatically to generate the drawings. - Your output should be conversational, interspersed with brief drawing instructions. Avoid drawing too much at once. To solve any given problem, you must draw at least twice. - If solving a problem involving X or Y axes, you must draw auxiliary lines on the X or Y axis to locate the target. - If the series label is shown, you do not need to align the value to the coordinate axes to obtain the number. If the label is not shown, you can align it to the coordinate axis or infer the value from other evidence. - The output format should be similar to: *"First, I will circle xx... BEGIN ... END. Hmm, it looks like I have drawn... Then I will... BEGIN ... END."* After each drawing, act as if you can visually interpret the content you have drawn to make the explanation vivid. - You must not use exhaustive methods for drawing. Draw only what is relevant to the question. For instance, if the X-axis line is missing, you need to infer the content. Partial conclusions must be derived through drawing.
Specific instructions are as follows: 1. Check and output whether all the data points involved have `series label show` information. - If `series label show=True`, there is no need to align the data points to the numerical axis; simply state the values based on the series label position. - If `series label show=False`, draw auxiliary lines to align the data points to the numerical axis for accurate evaluation.
2. There is no legend area annotation, so do not use `rectangle` to draw the legend area. Instead, use colors to describe the legend. 3. Do not reveal that you can see annotations or metadata.
Your output should be a string that simulates a human reasoning dialogue, with no additional content.

---

For reflective Sketch-CoT, we have built upon the non-reflective Sketch-CoT by introducing modifications. Through a carefully designed prompt for Qwen2.5-32B, we enable the model to randomly introduce errors during the reasoning process and subsequently self-correct them. The prompt ensures that errors are systematically generated and resolved, fostering a reflective problem-solving approach. The full content of the prompt can be found in Prompt C.

**Prompt for Reflective Sketch-CoT**

I need you to modify and refine the following dialogue by injecting reflective processes, replacing the originally completely correct solution process.
The method is as follows: Using BEGIN and END as boundaries, everything between and including END and what comes before it is called the "former part," and everything after END is called the "latter part." To create reflection, you first need to introduce an error.
The error type should be: coordinate errors between BEGIN and END in the former part, replacing them with coordinates of other points. Subsequently, you should correct the error by discovering and fixing it in the latter part, outputting new BEGIN and END commands to complete the dialogue.
Keep the language fluent and conversational. Let me further explain: The content between BEGIN and END is an operation, and you will "see" the result of the operation after END. Therefore, your reflection process must occur immediately after the END output, not after the erroneous reasoning concludes.
Note that there is no legend area; if you see one, you should remove it and use colors to describe the legend instead. Do not include any comments in the instructions, as the instruction code does not support comments. Do not include any hints that you are deliberately making errors.
The reflection must be brief and accurate, keeping the dialogue concise and organized. You should directly modify the dialogue rather than reflecting after erroneous reasoning ends. For example: ...BEGIN instruction END Wait/Hold on/Oh/Hmm, I might have made a mistake... (explain the reason for the mistake)... I'll redraw it now. BEGIN instruction END...
Here is the dialogue you need to modify:

# D    Detailed Sketch-MCTS Algorithm

Algorithm 1 demonstrates the detailed workflow of Sketch-MCTS, along with comprehensive descriptions of its specific parameters. Our proposed Sketch-MCTS represents a variant of the Monte Carlo Tree Search methodology. We have modified the termination condition of MCTS to conclude when either $SUCC_lim$ is satisfied or the number of simulations exceeds $SIM_lim$. Additionally, we have retained the low-temperature rollout approach to evaluate the value of the current node.

Compared to methods that randomly generate $N$ correct solution paths, our Sketch-MCTS offers the following advantages: First, we can evaluate the value of the current step in real-time, enabling stable expansion of the reasoning tree, whereas random rejection sampling methods of $N$ paths are entirely stochastic. Second, we can derive multiple correct and incorrect nodes from high-value nodes, creating step-level contrasts between correct and incorrect samples, which is crucial for off-policy reinforcement learning. In contrast, rejection sampling methods, with their completely independent reasoning paths between samples, cannot achieve step-level comparative analysis. Third, we can control the length of reasoning paths during the dynamic sampling tree process and ensure that the computational overhead of the sampling process remains manageable. Rejection sampling methods are static processes and cannot effectively control computational costs or reasoning path lengths.

| Symbol | Meaning | Default value |
|---|---|---|
| $q$ | textual query posed by the user | N/A |
| $I$ | chart image provided to the model | N/A |
| $y$ | gold (reference) answer | N/A |
| $\mathcal{L}$ | multimodal large language model queried during search | N/A |
| $\mathcal{R}$ | differentiable sketch renderer (ChartSketcher back-end) | N/A |
| $u.Q$ / $u.N$ | cumulative reward / visit count of node $u$ | 0 / 0 |
| $C_{\text{PUCT}}$ | exploration constant in the PUCT formulation | 1.9 |
| $\lambda_{\text{len}}$ | weighting factor for the depth-based penalty | 0.05 |
| $\varepsilon$ | small constant preventing division by zero | $1 \times 10^{-8}$ |
| SIM_lim | maximal number of MCTS iterations | 25 |
| SUCC_lim | early-stop threshold on successful terminal nodes | 3 |
| MAX_DEPTH | maximal dialogue depth (assistant–user turns) | 8 |
| $C_{\text{max}}$ | maximal expansions sampled per node | 6 |
| MAX_CHILD | maximal non-virtual children per node | 3 |
| $T_{\text{high}}/T_{\text{low}}$ | sampling temperatures for expansion / rollout | 0.9 / 0.4 |

---

**Algorithm 1** Sketch-MCTS

---

**Require:** visual query $q$, chart image $I$, ground-truth answer $y$, multimodal LLM $\mathcal{L}$, sketch executor $\mathcal{R}$
1: hyper–parameters $\Theta = \{\text{SIM\_lim}, \text{SUCC\_lim}, \text{MAX\_DEPTH}, C_{\max}, \text{MAX\_CHILD},$
2: $\quad\quad T_{\text{high}}, T_{\text{low}}, C_{\text{PUCT}}, \lambda_{\text{len}}, \varepsilon\}$
3: **function** UCB($u$) $\qquad\qquad\qquad\qquad\qquad\qquad\qquad$ ▷ depth-aware upper confidence bound
4: $\quad$ **return** $\underbrace{\dfrac{u.Q}{u.N + \varepsilon}}_{\text{exploitation}} + \underbrace{C_{\text{PUCT}}\sqrt{\dfrac{\ln(u.\text{parent}.N + 1)}{u.N + \varepsilon}}}_{\text{exploration}} - \underbrace{\lambda_{\text{len}}\Big(0.01\, u.\text{depth} + 0.3\big(e^{0.7\max(0,\, u.\text{depth}-4)} - 1\big)\Big)}_{\text{length penalty}}$

5: **end function**
6: root $\leftarrow$ NODE.INIT($[\text{user} : (q, I)]$)
7: successes $\leftarrow 0$
8: **for** $k \leftarrow 1$ **to** SIM\_lim **while** successes $<$ SUCC\_lim **do**
$\quad$ **Selection**
9: $\quad$ $v \leftarrow$ descendant of root maximising UCB($\cdot$)
$\quad$ **Expansion**
10: $\quad$ **if** $\neg v.\text{terminal} \wedge \neg v.\text{full} \wedge v.\text{depth} < \text{MAX\_DEPTH}$ **then**
11: $\quad\quad$ sample $\leq C_{\max}$ replies $\{r_i\} \sim \mathcal{L}(T = T_{\text{high}})$
12: $\quad\quad$ **for all** $r_i$ **do**
13: $\quad\quad\quad$ append $r_i$ to dialogue; extract SKETCH-COT program
14: $\quad\quad\quad$ **if** no code **then**
15: $\quad\quad\quad\quad$ add terminal child $u$; $u.\text{reward} \leftarrow$ ISRIGHT($r_i, y$)
16: $\quad\quad\quad$ **else**
17: $\quad\quad\quad\quad$ parse & render via $\mathcal{R}$; **on** failure **add** *virtual* child ($r = 0$) **continue**
18: $\quad\quad\quad\quad$ persist bitmap; send as user visual feedback
19: $\quad\quad\quad\quad$ duplicate/subset detection $\rightarrow$ virtualise redundant children
20: $\quad\quad\quad$ **end if**
21: $\quad\quad\quad$ **if** $u.\text{terminal} \wedge u.\text{reward} = 1$ **then** successes++
22: $\quad\quad\quad$ **end if**
23: $\quad\quad$ **end for**
24: $\quad\quad$ $v.\text{full} \leftarrow (\#\text{children} = \text{MAX\_CHILD})$
25: $\quad$ **end if**
$\quad$ **Rollout**
26: $\quad$ **if** $\neg v.\text{terminal} \wedge \neg v.\text{rolled} \wedge \neg v.\text{virtual}$ **then**
27: $\quad\quad$ simulate assistant $\rightarrow$ render $\rightarrow$ feedback with $T = T_{\text{low}}$
28: $\quad\quad$ terminate on no-code, render-fail, or depth limit; set $v.\text{reward}$
29: $\quad\quad$ $v.\text{rolled} \leftarrow$ true
30: $\quad$ **end if**
$\quad$ **Backpropagation**
31: $\quad$ **for** $u \in \text{ancestors}(v)$ with $\neg u.\text{virtual}$ **do**
32: $\quad\quad$ $u.N \leftarrow u.N + 1; \quad u.Q \leftarrow u.Q + v.\text{reward}$
33: $\quad$ **end for**
34: **end for**
35: **return** non-virtual child of root with maximal mean value $Q/N$

---

# E   Detailed experimental results

We conducted a comprehensive evaluation of ChartSketcher's chart comprehension ability and overall performance on 18 datasets. As shown in Table 8, VisCoT represents the weighted average of the remaining 12 general-purpose datasets, serving as an aggregated metric to assess the model's general understanding capabilities.

To further validate the robustness and generalization capabilities of our proposed ChartSketcher method, we conducted additional experiments on the challenging ChartX dataset. ChartX serves as a diverse benchmark for chart understanding, featuring more complex chart types and reasoning tasks. As presented in Table 3, we compared ChartSketcher against another advanced chart understanding model, ChartVLM.

## E.1   Comparison with ReFocus

In addition to evaluating on diverse benchmarks, we also conduct a direct performance comparison with *ReFocus* [10]. To ensure a fair and direct comparison, we evaluated our method on the test set provided by the ReFocus paper.

Table 3: Performance comparison on the ChartX benchmark. This experiment is conducted to further validate the generalization ability of our method on more diverse and complex chart datasets. SE denotes Structural Extraction with different tolerance levels.

| Model | SE (Structural Extraction) | | | QA |
| --- | --- | --- | --- | --- |
| | Strict | Slight | High | |
| ChartVLM | 23.18 | 30.68 | 38.30 | 43.84 |
| **ChartSketcher** | **35.65** | **41.74** | **47.06** | **56.94** |

Table 4: Performance comparison with ReFocus on its own test set (subsets of ChartQA). All models were evaluated under the same conditions to ensure a fair comparison.

| Model | chartqa-vbar | chartqa-hbar |
| --- | --- | --- |
| *Models based on Qwen2VL-2B* | | |
| Qwen2VL-2B (Base) | 51.31 | 48.65 |
| ChartSketcher-2B | 59.69 | 54.73 |
| *Models based on phi-3.5-4.2B* | | |
| phi-3.5-4.2B (Base) | 63.10 | 60.10 |
| ReFocus-phi3.5-4.2B | 72.20 | 71.00 |
| **ChartSketcher-phi3.5-4.2B** | **74.35** | **72.30** |

## E.2 Ablation on RL Sampling Strategy

To isolate and quantify the contribution of our proposed `Sketch-MCTS` algorithm, we conducted an additional ablation study. A key question raised during the review process was how our sophisticated Monte Carlo Tree Search-based sampling strategy compares against a simpler, more "naive" reinforcement learning approach. To address this, we designed a baseline that uses a simple rejection sampling method for collecting training data for the RL phase. This baseline, termed "w/ naive RL", embodies a less guided exploration strategy.

We trained a ChartSketcher-2B model with this naive RL data and compared its performance against our full model, which leverages Sketch-MCTS. The results are presented in Table 5.

Table 5: Ablation study on the RL sampling strategy. We compare the performance of ChartSketcher-2B trained with our proposed Sketch-MCTS against a version trained with a naive rejection sampling method ("w/ naive RL").

| Method | ChartQA-H | ChartQA-A | VisualCoT |
| --- | --- | --- | --- |
| ChartSketcher-2B (with Sketch-MCTS) | **55.60** | **80.88** | **66.86** |
| w/ naive RL | 52.08 | 78.24 | 62.12 |

## E.3 Analysis of Inference Efficiency

A practical consideration for any reasoning model is its inference efficiency and its handling of finite context windows. In this section, we provide a quantitative analysis of ChartSketcher's inference throughput.

**Inference Throughput.** Each step in the Sketch-CoT process requires the model to re-process the visual feedback, which makes its inference profile distinct from models that perform purely textual reasoning. To quantify this, we measured the inference throughput of ChartSketcher-72B and compared it against QvQ-Preview-72B and the base Qwen2VL-72B model. All experiments were conducted using the vLLM framework on a server with 8x A800 40G GPUs.

## E.4 Comparison with Chart-Expert Models

To provide a comprehensive view of ChartSketcher's performance within the specialized field of chart understanding, we compare our model with several prominent chart-expert models. These models are specifically designed and trained for chart-related tasks, making them highly relevant baselines. For this comparison, we use the widely-recognized ChartQA benchmark, which includes both machine-augmented (ChartQA-A) and human-annotated (ChartQA-H) question-answering sets. The results presented in Table 7 offer a nuanced perspective on ChartSketcher's capabilities.

Table 6: Inference throughput comparison. We report the average tokens per second for both the input processing and output generation stages. The overall speed is normalized relative to ChartSketcher.

| Model | Input Throughput (token/s) | Output Throughput (token/s) | Overall Speed (Normalized) |
|---|---|---|---|
| ChartSketcher-72B | **2995** | 329 | 1.00 |
| QvQ-Preview-72B | 570 | **1705** | 1.33 |
| Qwen2VL-72B | 1926 | 685 | 9.11 |

Table 7: Performance comparison with specialized chart-expert models on the ChartQA benchmark.

| Model | ChartQA-Augmented | ChartQA-Human |
|---|---|---|
| OneChart | 85.30 | 49.10 |
| ChartAst | **93.90** | 65.90 |
| **ChartSketcher-72B** | 92.64 | **85.20** |

Table 8: Comprehensive results on 18 benchmarks.

(a) Chart understanding Expert Benchmarks

| Model | ChartQA-H [30] | ChartQA-A | EvoChart-QA [15] | ChartBench [56] | PlotQA [36] | VisCoT [41] |
|---|---|---|---|---|---|---|
| *Proprietary models* | | | | | | |
| GPT-4o | 84.32 | 88.48 | 52.80 | 61.47 | 42.96 | 78.45 |
| Gemini-2.0 | 84.00 | 88.24 | 64.64 | 55.63 | 63.36 | 77.90 |
| Claude-3.5 | 85.04 | 90.72 | 56.96 | 57.63 | 60.64 | 75.93 |
| *Open-source / expert models* | | | | | | |
| Qwen2VL-72B | 82.48 | 88.56 | 54.00 | 54.77 | 73.76 | 72.14 |
| QvQ-Preview-72B | 83.20 | 89.76 | 54.32 | 42.40 | 69.04 | 76.52 |
| InternVL2.5-78B | 78.48 | 89.44 | 57.44 | 65.57 | 57.20 | 78.93 |
| ChartGemma-2B | 53.44 | 86.64 | 36.08 | 23.87 | 25.76 | 55.62 |
| Qwen2VL-2B | 50.48 | 75.84 | 23.84 | 20.27 | 38.80 | 58.33 |
| ChartSketcher-2B | 55.60 | 80.88 | 26.72 | 30.10 | 41.12 | 66.86 |
| ChartSketcher-72B | 85.20 | 92.64 | 63.28 | 68.33 | 76.72 | 76.59 |
| *Ablation (ChartSketcher-72B)* | | | | | | |
| w/o Rethink&RL | 77.76 | 91.12 | 51.12 | 50.40 | 67.76 | 72.58 |
| w/ Rethink w/o RL | 76.64 | 90.56 | 51.36 | 52.93 | 67.68 | 70.95 |
| w/o RL | 77.12 | 88.80 | 39.84 | 52.73 | 67.84 | 68.89 |
| w/o Feedback | 81.52 | 91.04 | 57.76 | 56.13 | 72.24 | 75.18 |
| w/o CoT | 75.12 | 90.08 | 55.36 | 47.43 | 68.16 | 76.12 |

(b) Generic / Document / Scene Benchmarks – Part A

| Model | OpenImages [20] | Flickr30k [39] | DocVQA [34] | CUB [49] | DUDE [21] | GQA [17] |
|---|---|---|---|---|---|---|
| *Proprietary models* | | | | | | |
| GPT-4o | 52.49 | 79.04 | 94.93 | 84.76 | 83.25 | 68.30 |
| Gemini-2.0 | 57.78 | 79.34 | 95.27 | 82.72 | 82.09 | 68.51 |
| Claude-3.5 | 62.50 | 75.68 | 97.64 | 70.93 | 84.40 | 60.63 |
| *Open-source / expert models* | | | | | | |
| Qwen2VL-72B | 51.75 | 61.64 | 93.36 | 71.14 | 82.75 | 57.06 |
| QvQ-Preview-72B | 60.21 | 72.96 | 92.68 | 74.19 | 84.41 | 63.91 |
| InternVL2.5-78B | 60.85 | 76.39 | 95.16 | 81.10 | 83.25 | 72.19 |
| ChartGemma-2B | 49.21 | 57.37 | 57.32 | 50.61 | 47.76 | 51.33 |
| Qwen2VL-2B | 53.97 | 34.48 | 79.50 | 50.20 | 71.31 | 23.31 |
| ChartSketcher-2B | 64.23 | 68.95 | 69.82 | 52.64 | 60.20 | 61.25 |
| ChartSketcher-72B | 68.68 | 72.19 | 92.68 | 68.09 | 79.10 | 65.85 |
| *Ablation (ChartSketcher-72B)* | | | | | | |
| w/o Rethink&RL | 62.33 | 66.04 | 89.08 | 71.14 | 82.75 | 62.17 |
| w/ Rethink w/o RL | 59.05 | 66.17 | 88.40 | 60.57 | 72.04 | 61.96 |
| w/o RL | 57.57 | 66.43 | 86.94 | 60.16 | 69.98 | 57.87 |
| w/o Feedback | 67.94 | 70.96 | 92.57 | 66.87 | 78.11 | 65.54 |
| w/o CoT | 62.43 | 69.53 | 92.23 | 77.64 | 78.28 | 62.88 |

(c) Generic / Document / Scene Benchmarks – Part B

| Model | TextVQA [43] | TextCap [42] | SROIE [16] | Infographic [33] | Emotic [19] | Visual7W [64] |
|---|---|---|---|---|---|---|
| *Proprietary models* | | | | | | |
| GPT-4o | 93.73 | 89.45 | 94.75 | 82.22 | 53.81 | 77.60 |
| Gemini-2.0 | 91.44 | 89.57 | 93.73 | 84.72 | 41.22 | 77.20 |
| Claude-3.5 | 94.30 | 89.10 | 95.15 | 78.26 | 35.42 | 73.70 |
| *Open-source / expert models* | | | | | | |
| Qwen2VL-72B | 94.49 | 87.81 | 94.31 | 78.89 | 43.14 | 73.90 |
| QvQ-Preview-72B | 89.35 | 84.76 | 94.61 | 84.72 | 65.55 | 69.90 |
| InternVL2.5-78B | 90.30 | 88.75 | 93.88 | 81.11 | 61.59 | 74.40 |
| ChartGemma-2B | 74.33 | 70.81 | 51.17 | 37.22 | 53.20 | 57.30 |
| Qwen2VL-2B | 88.97 | 80.77 | 94.02 | 47.78 | 50.15 | 60.40 |
| ChartSketcher-2B | 84.03 | 80.19 | 81.34 | 38.06 | 59.45 | 64.90 |
| ChartSketcher-72B | 90.11 | 85.58 | 89.65 | 74.44 | 67.16 | 73.00 |
| *Ablation (ChartSketcher-72B)* | | | | | | |
| w/o Rethink&RL | 86.50 | 82.42 | 88.34 | 70.83 | 58.54 | 65.80 |
| w/ Rethink w/o RL | 85.55 | 81.91 | 88.19 | 73.06 | 58.38 | 66.00 |
| w/o RL | 84.22 | 80.77 | 85.42 | 68.33 | 53.51 | 62.50 |
| w/o Feedback | 91.63 | 85.81 | 89.94 | 73.89 | 54.88 | 70.80 |
| w/o CoT | 90.49 | 85.70 | 94.90 | 78.89 | 64.70 | 67.20 |

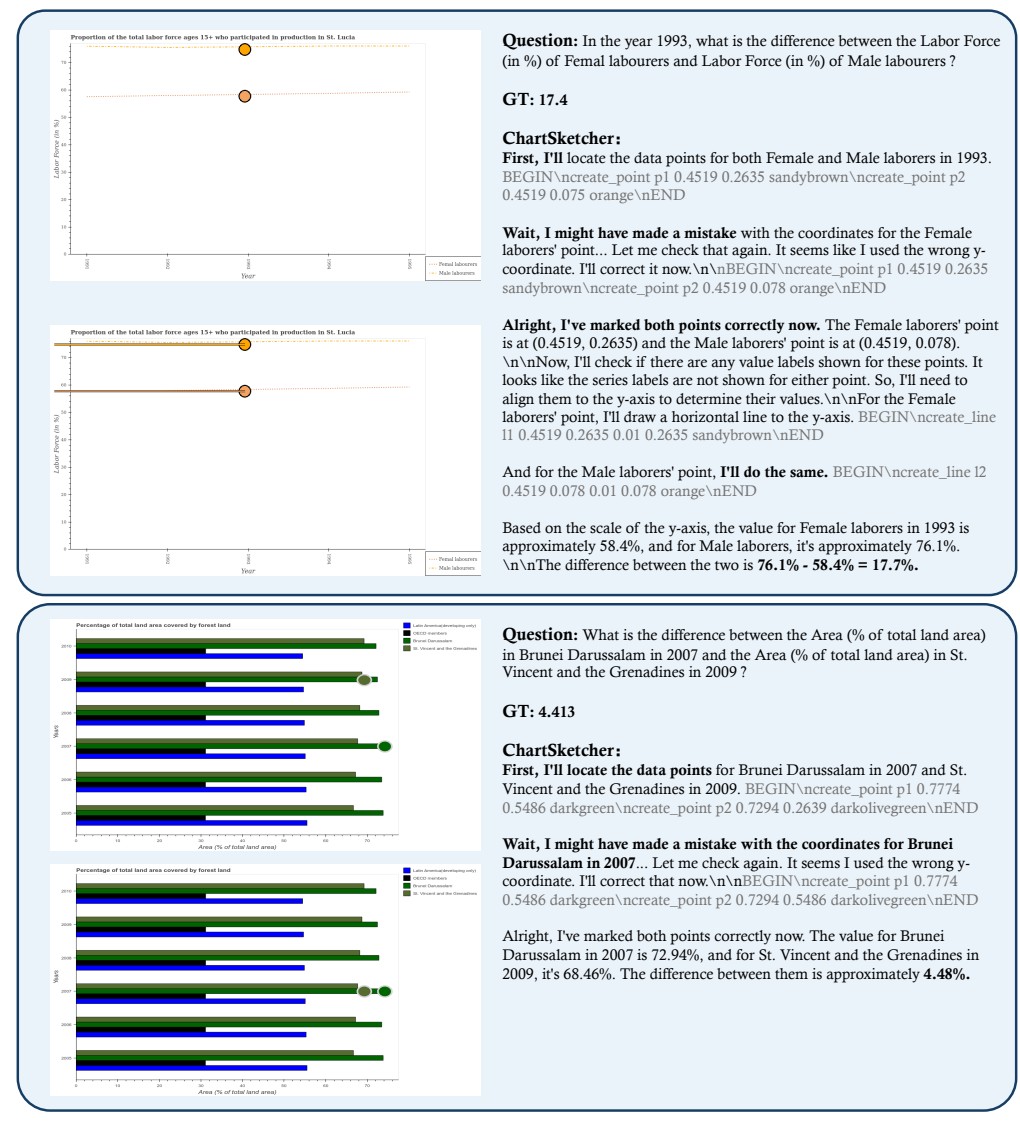

Figure 7: Visualization of an additional case study on PlotQA.

# F   Additional Cases of ChartSketcher

In this appendix, we provide a detailed account of the cases involving ChartSketcher across multiple datasets. As shown in Figure 7, Figure 8, and Figure 9, our approach offers rich user interactivity and explicit interpretability. Through the reinforcement learning process, ChartSketcher acquires reasoning capabilities that are absent during the cold start phase, demonstrating more conversational expressions compared to those in the cold start phase.

It is worth noting that, like many reasoning models, our approach faces common limitations such as ineffective self-reflection, overthinking, and infinite loops of reflection. To mitigate these issues, we lower the temperature during reasoning and cap the maximum chain length at 12. Addressing these limitations will be a key focus of our future work.

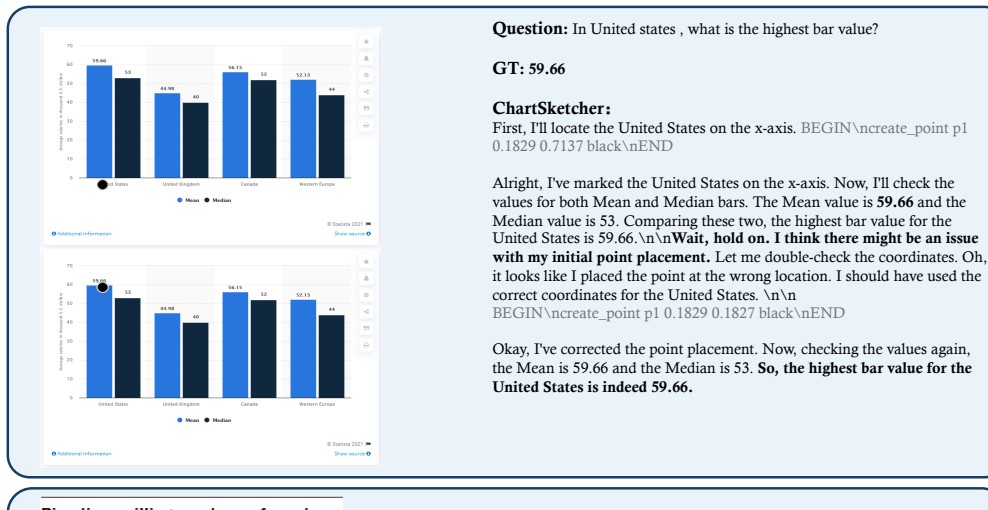

**Question:** In United states , what is the highest bar value?

**GT: 59.66**

**ChartSketcher：**

First, I'll locate the United States on the x-axis. BEGIN\ncreate_point p1 0.1829 0.7137 black\nEND

Alright, I've marked the United States on the x-axis. Now, I'll check the values for both Mean and Median bars. The Mean value is **59.66** and the Median value is 53. Comparing these two, the highest bar value for the United States is 59.66.\n\n**Wait, hold on. I think there might be an issue with my initial point placement.** Let me double-check the coordinates. Oh, it looks like I placed the point at the wrong location. I should have used the correct coordinates for the United States. \n\n BEGIN\ncreate_point p1 0.1829 0.1827 black\nEND

Okay, I've corrected the point placement. Now, checking the values again, the Mean is 59.66 and the Median is 53. **So, the highest bar value for the United States is indeed 59.66.**

---

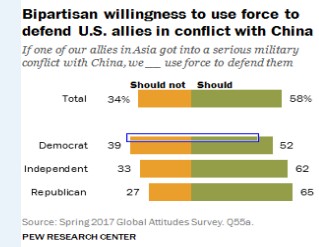

**Question:** Look at Democrats, Find the ratio between the two options?

**GT: 0.75**

**ChartSketcher：**

First, I'll locate the section for Democrats on the chart. BEGIN\ncreate_rectangle r1 0.349 0.529 0.784 0.562 blue\nEND

Alright, I've highlighted the Democrats' section. Now, I'll check the values associated with "Should" and "Should not" options. It looks like the value for "Should" is 52% and for "Should not" is 39%. The ratio between "Should" and "Should not" is approximately 1.33:1.

---

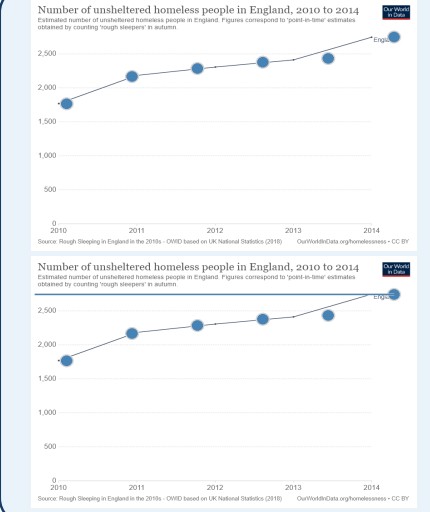

**Question:** When does the line reach the peak?

**GT: 2014**

**ChartSketcher：**

First, I'll locate the data points for England on the chart. BEGIN\ncreate_point p1 0.0927 0.4155 steelblue\ncreate_point p2 0.2622 0.3069 steelblue\ncreate_point p3 0.4316 0.2754 steelblue\ncreate_point p4 0.6011 0.2503 steelblue\ncreate_point p5 0.7705 0.2352 steelblue\ncreate_point p6 0.94 0.15 steelblue\nEND

Hmm, I might have made a mistake with the coordinates for the last point. Let me double-check that. Oh right, the correct y-coordinate should be 0.15 instead of 0.15. I'll fix this now.\n\n BEGIN\ncreate_point p1 0.0927 0.4155 steelblue\ncreate_point p2 0.2622 0.3069 steelblue\ncreate_point p3 0.4316 0.2754 steelblue\ncreate_point p4 0.6011 0.2503 steelblue\ncreate_point p5 0.7705 0.2352 steelblue\ncreate_point p6 0.94 0.152 steelblue\nEND

Alright, I've marked all the data points for England. Now, I'll check if there's a value label showing the values. It looks like the series label is not shown, so I'll need to align the data points to the chart's data axis to determine their values. \n\nFor the point in 2014, I'll draw a horizontal line to the y-axis. BEGIN\ncreate_line l1 0.94 0.152 0.01 0.152 steelblue\nEND

Based on the scale of the y-axis, it can be seen that the value for England in 2014 is about 2740. This is the highest value among all the data points, indicating that the line reaches its peak in **2014**.

Figure 8: Visualization of an additional case study on ChartQA.

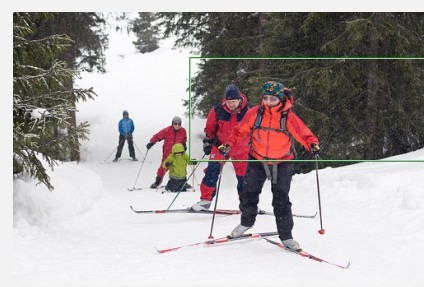

**Question:** What's around the man?
**GT: trees**

**ChartSketcher:**
First, I'll locate the area around the man in the image. BEGIN create_rectangle r1 0.428 0.168 1.000 0.546 green END

Alright, I've highlighted the area around the man. It looks like there are trees in this region. According to the visual information, the man is surrounded by **trees**.

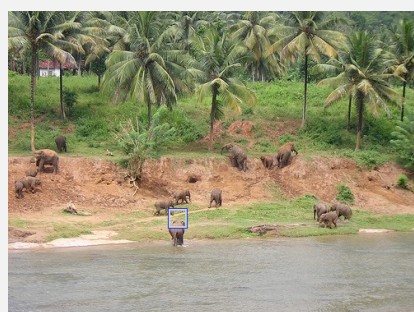

**Question:** What is the animal that is in the water called?

**GT: elephant**

**ChartSketcher:**
First, I'll locate the area where the animal is positioned in the water. BEGIN create_rectangle r1 0.388 0.642 0.434 0.706 blue END

Alright, I've highlighted the region where the animal is located. Based on the visual information, it looks like the animal in the water is an **elephant**.

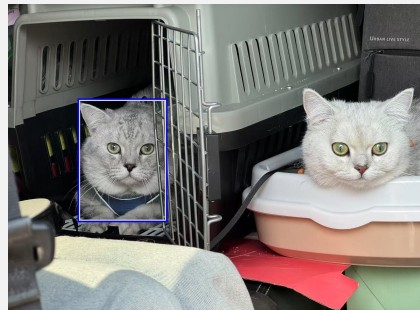

**Question:** Where is the deep grey cat?

**GT: In the cage**

**ChartSketcher:**
First, I'll locate the area where the deep grey cat might be. BEGIN create_rectangle r1 0.169 0.306 0.387 0.706 blue END

Alright, I've highlighted the area where the deep grey cat is located. Based on the visual information, **it looks like the cat is sitting inside a cage.**

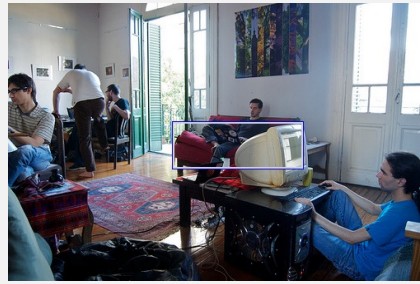

**Question:** Which kind of furniture is not wooden?

**GT: couch**

**ChartSketcher:**
First, I'll locate the area where the furniture is positioned in the image. BEGIN create_rectangle r1 0.398 0.426 0.716 0.598 blue END

Alright, I've highlighted the area where the furniture is positioned. Now, let's check if there's a label showing the type of furniture. It looks like there's a label showing the value inside the sector. According to the label, the value associated with this furniture is a **sofa**.

Figure 9: Visualization of an additional case study on general image.

