# OpenReview forum: "ChartSketcher: Reasoning with Multimodal Feedback and Reflection for Chart Understanding"
_NeurIPS.cc/2025/Conference — NeurIPS 2025 poster_

### Official Review · Reviewer_6CLL · 2025-06-30

**Clarity:** 2
**Significance:** 4
**Originality:** 3
**Rating:** 4
**Confidence:** 4

**Summary:**

This paper introduces ChartSketcher, a multimodal feedback-driven stepwise reasoning method. Through Sketch-CoT, the model annotates intermediate reasoning steps on charts using a programmatic plotting library, and iteratively feeds visual annotations back into the reasoning process to achieve visual anchoring and progressive optimization. The model employs a two-phase training strategy of "cold-start learning + offline reinforcement learning." Experiments show that it excels in chart comprehension benchmarks and general visual tasks, providing an interactive and interpretable solution for chart understanding.

**Questions:**

1. How reliant is the proposed method on the model's coordinate localization ability? If the baseline's localization or coordinate output is weak, will the method still work?
2. What does the reasoning process look like for cases with 6-7 steps? Could the model response fall into a loop?

**Ethical Concerns:**

["NO or VERY MINOR ethics concerns only"]

**Final Justification:**

Thanks to the authors for their response, which addressed most of my concerns. I would like to give a positive score if the authors can improve their paper writing about the SFT and RL workflow.

**Limitations:**

yes

**Paper Formatting Concerns:**

The reviewer suggested that some key insights be expressed precisely in formulas. Not even a single formula appears in the entire paper, which poses a challenge to understand the practices and contributions of the paper.

**Quality:**

3

**Strengths And Weaknesses:**

**Pros:**

1. The motivation is sound, using tools to mimic human sketch-based understanding is an interesting direction in VLM RL.
2. The authors provide detailed descriptions of implementation, including prompts, commands, and algorithms.
3. Experiments show that ChartSketcher performs exceptionally well.

**Cons:**

1. The core idea is unclear (Line 124-129). Reviewer is hard to intuitively understand the intermediate steps of reasoning. How visualisations interact with the model during cold-start and RL training? Are only instructions command code used, or do visual tokens get filtered and re-encoded via ViT for the token sequence?
2. There is no analysis of inference speed or generative token length. Will the model's maximum context length limit the rethink mechanism?
3. There is a lack of training data visualization, raising concerns about data quality and potential homogeneity.

---

> ### Author Rebuttal · Authors · 2025-07-31
>
> ## overall
> We would like to express our gratitude for your thorough and insightful review. Your feedback has been invaluable for improving our explanations and clarifying our contributions. We have carefully considered each of your points and appreciate the opportunity to provide a more detailed clarification of our work.
>
> ## cons1:
> ## reply:
> We apologize if our description was not clear enough.
>
> To be explicit, first, at every reasoning step, the generated sketch visualization is fully re-encoded by the model's ViT.
>
> ChartSketcher's interaction is a multi-turn, multi-modal dialogue. For a single reasoning step, the precise data flow is as follows:
>
> * **1.	Input (Turn $t$):** The MLLM receives the current image $Image_t$ and text prompt $Text_t$ (which includes the dialogue history).
> * **2.	Generation:** The model's language decoder generates a response containing reasoning text and a `<code>` block with sketch instructions.
> * **3.	Rendering:** Our system's backend renderer receives $Image_t$ and the sketch instructions from `<code>`, and generates a new image, $Image_{t+1}$, with the corresponding sketch already drawn on it.
> * **4.	Feedback ($Turn  t+1$):** This new image, $Image_{t+1}$, will become the visual input for the next turn. It is fed into the model's ViT to generate new visual tokens. These visual tokens, along with the updated text prompt, will form the input for the next reasoning step.
>
> When the model believes the reasoning is complete, it no longer outputs `<code>`, and thus the reasoning loop ends. This ultimately forms a sequence of `<image><query> <think1><code> <feedback image 1> <think2><code> <feedback image 2> ...... <final think>`.
>
> Therefore, the model is not merely "reading" the code it generated. It is visually perceiving the consequences of its actions by processing the newly rendered image at each step. We will revise Section 3.1 of the paper to highlight this iterative "image-ViT-feedback" loop more clearly.
>
> ## cons2:
> ## reply:
> This is a very practical and valuable question. Similar to reasoning models like DeepSeek-R1, the maximum context length indeed limits the number of reflections the model can perform, and we have made trade-offs in our practical engineering implementation.
>
> **Inference Speed:** Each step in the Sketch-CoT chain requires reading the feedback image, which makes the inference speed slightly slower compared to general reasoning models. On average, across all datasets involved in testing, our inference speed per query is 75% of QvQ's. The specific inference throughput is shown in the table below. All experiments were conducted with vLLM on a machine with 8x A800 40G GPUs.
>
> This table clearly shows the model's throughput and also reveals interesting results. ChartSketcher, due to the need to check whether previous reasoning steps were correct, requires higher performance for input token prefill during inference. QvQ primarily focuses on text reasoning, thus requiring higher decode throughput. Qwen2VL-72B, without "Thinking," has significantly shorter output tokens than input tokens, so its input and output performance are relatively balanced, with the main bottleneck being input prefill.
>
> | Model | Inference Speed (Input) Average token/s | Inference Speed (Output) Average token/s | Overall Inference Speed (Average across all test sets) Normalized |
> |-|-|-|-|
> | ChartSketcher-72B | 2995 | 329 | 1 |
> | QvQ-72B | 570 | 1705 | 1.334 |
> | Qwen2VL-72B | 1926 | 685 | 9.107 |
>
> **Context Length Limit:** Yes, the model's maximum context length is indeed a hard constraint on the reflection mechanism. The entire dialogue history, including all text and <code> blocks, accumulates with each turn. For models with limited context windows, overly long reasoning chains can indeed lead to content truncation. To manage these limitations, we have implemented two key safeguards, which we will emphasize more clearly in the paper:
>
> * **1.Maximum Dialogue Depth:** As described in Appendix F, we set an upper limit for the maximum length of the reasoning chain (typically 8-12 steps) to prevent infinite loops and manage context length.
>
> * **2.Efficient Search via Sketch-MCTS:** Our Sketch-MCTS algorithm is designed to find the correct solution more efficiently than naive exploration. By pruning redundant and unpromising paths, it helps keep the reasoning chain concise, thereby saving both time and context window space. This RL strategy implicitly saves token count during the inference time.
>
> ## cons3:
> ## reply:
> We agree that visualizing training data is crucial for the transparency of our work. To address this, we will add a new dedicated section in the appendix for visualizing our synthetic data.
>
> Here, we detail the scale of the data we used. All data has been rigorously checked and will be open-sourced to reduce the potential for data homogeneity. As you can see, we used synthetic chart data along with VisualCoT and its annotations in the first ColdStart phase (without reflection). In the second ColdStart phase (with reflection), we exclusively used synthetic chart data. For the RL phase, we utilized real-world Chart datasets and VisualCoT's QA pairs. All data will be publicly accessed and open-sourced.
>
> | Training Phase | Method | Data Source                 | Data Type                 | Quantity         |
> | :- | :- | :- | :- | :- |
> | Cold Start     | SFT    | EvoChart Synthetic Chart Data | Correct Reasoning Path    | 155203 (87.3%)   |
> |                |        | VisualCoT and its Annotations | Correct Reasoning Path    | 22510 (12.7%)    |
> |                |        | Total                       |                           | 177713           |
> |                | DPO    | EvoChart Synthetic Chart Data | Reflection Reasoning Path | 147955           |
> | RL             | KTO    | ChartQA and ChartBench      | MCTS Sampled Paths        | 41196 (81.6%)    |
> |                |        | General QA-Pairs *          | MCTS Sampled Paths        | 9259 (18.4%)     |
> |                |        | Total                       |                           | 50455            |
>
>
> \*The datasets for the RL phase include TextVQA, TextCaps, DocVQA, DUDE, SROIE, CUB-200-2011, Flickr30k, Visual7W, InfographicsVQA, VSR, GQA, and OpenImages. The term "VisualCoT dataset" is, in fact, an aggregated name for the aforementioned datasets.
>
> ## question1:
> ## reply:
> Thank you for your question regarding the robustness of our work. While the strong initial localization capability of the base model is undoubtedly beneficial, ChartSketcher is specifically designed to be robust to, and even correct, its own localization errors.
>
> Here's why:
>
> * If the model's initial localization capability is weak, it might generate a sketch with incorrect coordinates.
> * However, in the very next step, the model will "see" its own mistake, because that incorrectly rendered sketch now becomes part of its visual input.
> * During the reflective training phase, the model learns to identify this inconsistency between its textual reasoning and the visual evidence. It learns to generate corrective dialogue.
> * This self-correction loop allows the model to iteratively refine its understanding and eventually converge to the correct answer.
>
> Therefore, our method does not solely rely on the base model's capabilities; it actively enhances and corrects them. In fact, the 2B version of ChartSketcher, trained on the Qwen2VL-2B which has weaker localization capabilities than 72B version, also achieved significant improvements. To further illustrate the correlation between the model's capabilities and our method, we have also trained a medium-sized 4.2B version of the model, which will be open-sourced soon and its performance added to the article's Appendix.
>
> ## question2:
> ## reply:
> This question is similar to cons 2, and it's a very practical one.
>
> First, allow me to answer: a 6-7 step reasoning process typically occurs for complex problems that require 2-3 sub-steps and may involve self-correction. An example might be as follows:
>
> Step 1: "First, I need to understand what the blue and orange lines represent. I'll highlight the legend." (Draws a box around the legend).
>
> Step 2: "Okay, blue is 'Good' and orange is 'Bad'. Now I need to find the value for 'Good' in 2015." "2015 is between 2014 and 2016. I will mark this position on the x-axis." (Draws a point on the x-axis).
>
> Step 3: "Now I will draw a vertical line until it intersects the blue line." (Draws a vertical line).
>
> Step 4 (Self-correction): "Wait, I accidentally drew the line to the orange line. Let me erase it and redraw it to the blue line." (Deletes the old line, draws a new correct line).
>
> Step 5: "Now I'll draw a horizontal line from this point to the y-axis to read the value." (Draws a horizontal line).
>
> Step 6: "This line intersects the y-axis at approximately 57. The answer is 57." (No Sketch, reasoning ends).
>
> Second, we will address your concern about infinite loops.
>
> This question is very insightful for Reasoning Models. Honestly, for all existing Reasoning Models, infinite loop reasoning is possible, which can lead to uncontrolled context. Our model is no exception, but we have designed a series of steps to prevent such occurrences.
>
> Additionally, we have implemented two mechanisms to prevent this from happening:
>
> **Maximum Depth Limit:** We enforce a hard upper limit on the number of reasoning steps.
>
> **MCTS - Shaping:** The reinforcement learning process implicitly discourages overly long or circular reasoning, as these paths are less likely to efficiently reach the correct answer and thus receive lower cumulative rewards.
>
> ## Paper Formatting Concerns:
> We apologize for the confusion we caused. The detailed algorithmic workflow of our work is presented in Appendix D. We will revise the main text to provide a more explicit and formalized description of the ChartSketcher workflow.

---

> > ### Comment · Reviewer_6CLL · 2025-08-06
> >
> > Thanks to the authors for their response, which addressed most of my concerns. I would like to give a positive score if the authors can improve their paper writing about the SFT and RL workflow.

---

> > > ### Author Response · Authors · 2025-08-06
> > >
> > > We sincerely appreciate you taking the time to engage with our rebuttal and for your recognition of our work. We will ensure that the writing in Section 3.1 3.2 and Appendix D is improved in the revised manuscript, as per your suggestion.

---

> ### Comment · Area_Chair_ZQXP · 2025-08-03
>
> Dear Reviewer,
>
> Could you please check if the authors’ rebuttal adequately addresses your concerns? If so, kindly acknowledge the rebuttal and provide any additional comments. If not, it would be greatly appreciated it if you could engage in a discussion with the authors. Your input at this stage is essential to the review process. Thank you very much for your time and effort!
>
> AC

---

### Official Review · Reviewer_z7tZ · 2025-07-01

**Clarity:** 3
**Significance:** 3
**Originality:** 3
**Rating:** 4
**Confidence:** 4

**Summary:**

In this paper, the authors introduce ChartSketcher, a novel multimodal reasoning framework that enhances chart understanding by enabling large language models to iteratively annotate and refine their reasoning steps directly on chart images. Drawing inspiration from human sketching behavior, the authors propose a Sketch-CoT approach where the model generates visual annotations (e.g., arrows, lines) to ground its reasoning, then uses programmatic sketching libraries to refine these annotations based on visual feedback. A two-stage training strategy combines cross-modal knowledge distillation (cold start) with off-policy reinforcement learning (RL) optimized via a customized MCTS algorithm (sketch-MCTS). Experiments show strong performance on chart-specific benchmarks (e.g., ChartQA, PlotQA) and general vision tasks (e.g., OpenImages), demonstrating improved accuracy and interpretability compared to baseline models like GPT-4o and Qwen2VL. While the technical contributions are solid and the experimental setup is thorough, the evaluation could be strengthened by broader dataset coverage (e.g., more diverse chart types/visual tasks) and deeper analysis of failure modes.

**Questions:**

1. Experiments focus heavily on chart-specific benchmarks (ChartQA, PlotQA) but lack diversity in visual task types (e.g., infographics, mixed-modal charts, 3D plots). How do you justify this scope, and what steps will you take to validate ChartSketcher’s robustness across diverse visual domains?

2. Ablation studies focus on component removal (e.g., "w/o Rethink&RL") but do not quantify the marginal benefit of key innovations like sketch-MCTS vs. naive RL. Can you redesign ablations to isolate the impact of iterative sketching/refinement vs. baseline MCTS?

3. ChartSketcher struggles with charts requiring 3D perspective (e.g., financial candlesticks) or dynamic updates (e.g., live dashboards). What architectural changes would be needed to handle such cases, and how would your training strategy adapt?

**Ethical Concerns:**

["NO or VERY MINOR ethics concerns only"]

**Limitations:**

Yes

**Quality:**

3

**Strengths And Weaknesses:**

​​Strengths​​:
For strength, I would like to say that this paper presents a technically sound and innovative approach to chart understanding by introducing ChartSketcher, a multimodal reasoning framework that bridges human-like sketching behavior with programmatic visual feedback:
(1) Sketch-CoT method enables iterative annotation and refinement of reasoning steps directly on chart images, addresses a critical gap in existing MLLMs that struggle with complex visual reasoning.
(2) Two-stage training strategy (cold-start knowledge distillation + RL optimization via sketch-MCTS) is well-designed, and the experimental results demonstrate strong performance gains on both chart-specific benchmarks (e.g., ChartQA, PlotQA) and general vision tasks (e.g., OpenImages).
(3) The work’s emphasis on interpretability and self-correction through visual grounding is particularly impactful, offering a practical pathway for improving automated data analysis tools.

​​Weaknesses​​:
Contributions are significant, however, the evaluation has limitations that temper the paper’s impact:
(1) Experiments prioritize chart-specific datasets (e.g., ChartQA) but lack diversity in visual task types (e.g., fewer tests on infographics or mixed-modal charts).
(2) Additionally, the theoretical analysis of why iterative sketching improves reasoning—beyond empirical validation—is sparse.

---

> ### Author Rebuttal · Authors · 2025-07-31
>
> ## overall
> We sincerely thank Reviewer for the insightful feedback on our work. We are encouraged that the reviewer recognizes the innovation of our ChartSketcher method, the well-designed two-stage training strategy, and the practical value of our work in enhancing model interpretability. We will address the identified weaknesses and questions below.
>
> ## Weakness1:
> ## reply:
> ### 1: Thank you for your valuable feedback. We are pleased that you recognize the value of our work. Regarding the diversity of our evaluation, we would like to clarify that we did make a conscious effort to validate ChartSketcher's robustness on a range of visual tasks beyond charts.
>
> As you can see in the appendix, our evaluation across 18 benchmarks already includes challenging datasets such as InfographicVQA and DocVQA. These datasets are characterized by complex, mixed-modal layouts with dense text and graphics, demanding a different and often more comprehensive reasoning capability than standard charts. Our strong performance on these benchmarks demonstrates that the core principles of ChartSketcher--iterative visual grounding and self-correction—can indeed generalize to more unstructured and diverse visual domains. To further illustrate the effectiveness of our method, we additionally tested the ChartX dataset, which we will add to the Appendix for further comparison.
>
> | Model | SE (strict) | SE (slight) | SE (high) | QA |
> | :--- | :--- | :--- | :--- | :--- |
> | **ChartSketcher** | 35.65 | 41.74 | 47.06 | 56.94 |
> | **ChartVLM** | 23.18 | 30.68 | 38.30 | 43.84 |
>
> ### 2:Thank you for this excellent and thought-provoking point. Let us address your concerns regarding theoretical analysis.
>
> As we replied to reviewer QsY7, theoretical analysis is indeed very important. Currently, many works[1][2][3] are inspired by human behavior. These works have empirically demonstrated effectiveness, but a theoretical analysis of human cognitive behavior is quite challenging. Our primary contribution in this work is ChartSketcher, which successfully operationalizes this human-like strategy and, in doing so, provides a highly effective and interactive reasoning experience. We see our work as the foundational step of validating that this approach works and works well in practice. We believe that a formal theoretical validation is necessary, but also a grand undertaking that could constitute a significant research program in its own right. Such an analysis might involve frameworks from information theory or formal methods.
>
> While a full theoretical treatment is beyond the scope of this paper, we can hypothesize about the underlying principles:
>
> Cognitive Offloading: Each sketch acts as an external memory, offloading the burden of holding complex spatial relationships and intermediate conclusions in the model's "working memory". This allows for more focused, incremental processing.
>
> Error Correction via Grounded Feedback: The visual feedback loop provides a powerful mechanism for error detection. By comparing its intended action with the actual outcome, the model gets a grounded signal to correct flawed internal states, which is much more direct than purely text-based reflection.
>
> We are grateful you raised this point. In our revised manuscript, we will add a discussion in the conclusion to better frame our contribution in this context and explicitly highlight the need for this future theoretical work, suggesting it as an exciting and important path forward for the community.
>
> [1]Mind's Eye of LLMs: Visualization-of-Thought Elicits Spatial Reasoning in Large Language Models [NeurIPS 2024]
>
> [2]Enhancing LLM Reasoning via Vision-Augmented Prompting [NeurIPS 2024]
>
> [3]Goal Reduction with Loop-Removal Accelerates RL and Models Human Brain Activity in Goal-Directed Learning [NeurIPS 2024]
>
> ## question1:
> ## reply:
> Thank you for this insightful question about the scope of our work.
>
> To clarify our evaluation scope, we did test ChartSketcher on 18 diverse benchmarks, which do include complex mixed-modal datasets like InfographicVQA. However, it is correct to point out that our evaluation has its limitations. We have not explicitly tested on 3D plots or dynamic video-based charts. We acknowledge this as a limitation of our current work's scope.
>
> We can demonstrate a clear and feasible path for ChartSketcher to address these domains, grounded in its core principles:
>
> 1.For any 2D visual representation, including mixed-modal charts and 2D projections of 3D plots: The fundamental act of sketching to deconstruct a problem remains viable. This mirrors the intuitive, tool-agnostic way humans approach complex visual information. For a 3D plot, for instance, our model could sketch on its 2D projection to highlight points, axes, or surfaces. The primary work would involve extending our programmatic library with primitives that are aware of the 3D context.
>
> 2.For dynamic content, videos: We can adapt our method by sampling keyframes. By treating a keyframe as a static image, Sketch-CoT can be directly applied to analyze a specific moment in time. This approach could even be extended to reason about changes by applying sketches to consecutive frames and prompting the model to compare them.
>
> In essence, while our current work focused on establishing a strong foundation on a wide array of static benchmarks, the ChartSketcher framework is inherently extensible due to its mimicry of a fundamental human problem-solving process.
>
> ## question2:
> ## reply:
> Thank you for your suggestion. We completely agree with your advice regarding our ablation studies. Comparing Sketch-MCTS with a general naive sampling method is necessary to demonstrate the effectiveness of Sketch-MCTS.
>
> Therefore, we have added an ablation experiment to prove the effectiveness of Sketch-MCTS. We used a simple rejection sampling method to obtain data, embodying the characteristics of "naive RL." The experimental results are as follows. It can be seen that, compared to Sketch-MCTS, the simple rejection sampling method does not perform well. This is reasonable because simple rejection sampling algorithms cannot constrain the reasoning process within a concise and efficient range, leading to excessively lengthy reasoning chains.
>
> | method            | ChartQA-H | ChartQA-A | visCoT  |
> |-|-|-|-|
> | ChartSketcher-2B  | 55.60     | 80.88     | 66.86   |
> | w/ naive RL        | 52.08      | 78.24      | 62.12   |
>
> ## question3:
> ## reply:
> **1. Handling 3D Perspective Charts**
>
> Regarding 3D views, we concur with the implicit premise of your question: current MLLMs generally struggle with true 3D spatial understanding and depth perception. In practice, 3D information is almost always presented and processed as a 2D projection on a screen.
>
> Given this reality, our Sketch-CoT methodology remains directly applicable and highly relevant. The model would operate on the 2D flattened image of the 3D chart, using sketches to deconstruct the visual information presented within that projection. The core challenge is not about rendering in 3D, but about teaching the model to correctly interpret the visual cues of depth and perspective present in the 2D image.
>
> **2. Handling Interactive Info (Live Dashboards)**
>
> For interactive elements like dashboards, we believe the solution lies beyond simple sketching and requires evolving the model's action space. We envision extending our framework by incorporating key "point-and-click" and "drag-and-drop" operations.
>
> This evolution would position the resulting model as a powerful hybrid of a Sketching Reasoner and a GUI Agent. It would not only analyze static visual content through sketches but also actively interact with interface elements to reveal new information or trigger updates. For example, it could "click" on a dropdown menu to filter data or "drag" a time-slider to see how a chart evolves, then apply Sketch-CoT to the resulting view. This represents a fascinating and powerful fusion of passive reasoning and active interaction.
>
> **3. Adaptation of the Training Strategy**
>
> To train a model capable of such complex, multi-step interactions, a shift in our training strategy would indeed be necessary.
>
> We agree that moving from our current off-policy RL to an on-policy RL approach (like GRPO) would be more suitable. On-policy methods are better at exploring and learning in environments where the consequences of an action are complex and path-dependent.
>
> Furthermore, we would need to implement a step-level reward mechanism. Instead of only rewarding the final correct answer, we would provide intermediate rewards for successfully completing sub-tasks, for example, a positive reward for correctly clicking a filter button that reveals the target data. This dense reward signal is crucial for guiding the model through the long and complex action sequences required for effective Sketch navigation and analysis.

---

> > ### Comment · Reviewer_z7tZ · 2025-08-03
> >
> > Thanks for authors' responses and rebuttal. I think the score is appropriate for current version.

---

> > > ### Author Response · Authors · 2025-08-05
> > >
> > > We sincerely appreciate you engaging with our rebuttal. We have taken note of your final decision and are grateful for the conclusion of this discussion.

---

> ### Comment · Area_Chair_ZQXP · 2025-08-03
>
> Dear Reviewer,
>
> Could you please check if the authors’ rebuttal adequately addresses your concerns? If so, kindly acknowledge the rebuttal and provide any additional comments. If not, it would be greatly appreciated it if you could engage in a discussion with the authors. Your input at this stage is essential to the review process. Thank you very much for your time and effort!
>
> AC

---

### Official Review · Reviewer_hekG · 2025-07-03

**Clarity:** 2
**Significance:** 2
**Originality:** 2
**Rating:** 4
**Confidence:** 4

**Summary:**

This paper focuses on multi-step chart visual understanding. They propose ChartSketcher, a multimodal feedback-driven step-by-step reasoning method. ChartSketcher is a chart understanding model that employs Sketch-CoT, enabling MLLMs to annotate intermediate reasoning steps directly onto charts using a programmatic sketching library, iteratively feeding these visual annotations back into the reason14 ing process. This mechanism enables the model to visually ground its reasoning and refine its understanding over multiple steps. They employ a two-stage training strategy: a cold start phase to learn sketch-based reasoning patterns, followed by off-policy reinforcement learning to enhance reflection and generalization. Experiments demonstrate that ChartSketcher achieves promising performance.

**Questions:**

1. Could you add citation and comparison to the papers: [ReFocus: Visual Editing as a Chain of Thought for Structured Image Understanding] from ICML 2025 and [Visual Sketchpad: Sketching as a Visual Chain of Thought for Multimodal Language Models] from Neurips 2024?
2. What's the training data exact size for each different source at each stage?
3. What's the performance without using Visual-CoT at all?
3. What's the performance with only using Visual-CoT?

**Ethical Concerns:**

["NO or VERY MINOR ethics concerns only"]

**Final Justification:**

The rebuttal has answered all my questions and added the required experiments / details. I have increased my score correspondingly.

**Limitations:**

yes

**Quality:**

2

**Strengths And Weaknesses:**

Strengths:
1. Provides a huge high-quality chart reasoning dataset and could be helpful.
2. Shows efficacy of RL on chart reasoning problems.

Weakness:
1. Limited Novelty. There is a similar paper: [ReFocus: Visual Editing as a Chain of Thought for Structured Image Understanding] from ICML 2025. The idea seems same, and the experiment result of ReFocus (on Phi3.5vision) with ChartSketcher-2B seems similar. Please add comparison to it.
2. Lack Citation: [Visual Sketchpad: Sketching as a Visual Chain of Thought for Multimodal Language Models] from Neurips 2024. Also lack comparison to it.
3. The experiment needs to done more clearly, especially if you want to draw the conclusion that ChartSketcher proposes a chart reasoning dataset that helps chart and also helps general VQA tasks. In this case, First, the training data exact size for each different source at each stage should be listed. Second, since the Visual-CoT data was also used, it's hard to say that the model improves on general VQA tasks from Chart data. The paper should do the experiment with and without using Visual-CoT data to make more precise claims. Right now, if the proposed dataset is the biggest contribution, but the experiments are done with Visual-Cot, it's hard to tell whether the experiment conclusions can hold.
4. Continue from 3, the paper should also train a version with only Visual-CoT to compare with on general VQA tasks as a baseline.

---

> ### Author Rebuttal · Authors · 2025-07-31
>
> ## overall
> Thank you very much for your insightful and critical feedback on our paper. Your points regarding comparison with recent related work and clarification of the experimental data composition are valuable for improving the rigor and clarity of our paper. We will add citations to the works you mentioned and improve our work in the revised version. In the following response, we will elaborate on the uniqueness of our method and explain and clarify the data details and ablation studies you requested.
>
> ## weakness1,2 & question1:
> Thank you for pointing out these two valuable related works. We will add detailed citations and discussions in the revised version.
>
> To provide a direct comparison between Refocus and ChartSketcher, we present the performance comparison between ChartSketcher-2B and Refocus-phi3.5 on the Refocus test set. It can be noted that ChartSketcher's base model (Qwen2VL-2B) is significantly weaker than the base model of Refocus, phi-3.5 (4.2B). Therefore, to ensure a fair comparison, we have retrained a version of ChartSketcher based on phi-3.5 (which has now been open-sourced).
>
> | Model            | chartqa-vbar | chartqa-hbar |
> |----|--------------|--------------|
> | Qwen2VL-2B       | 51.31        | 48.65        |
> | ChartSketcher-2B | 59.69        | 54.73        |
> | ChartSketcher-phi3.5-4.2B | 74.35      | 72.30        |
> | phi3.5-4.2B           | 63.1         | 60.1         |
> | Refocus-phi3.5-4.2B   | 72.2         | 71.0         |
>
> The comparative experiments are also shown in the table above. This demonstrates the effectiveness and competitiveness of our method. All experiments were run on four 8 x A800 (40GB) servers.
>
> ### **Regarding the comparison to Sketchpad and Refocus**, we would like to elaborate on how our work builds upon and diverges from these approaches.
>
> First, ChartSketcher differs from the works you mentioned in its domain. ChartSketcher primarily focuses on charts—such as bar, line, and pie charts—and our solution can also be applied to natural images or structured images, like documents. The two works you mentioned, such as Refocus, mainly focus on structured data like tables and charts(only include bar chart in train/test set).
>
> Second, ChartSketcher differs from the works you mentioned in its methodology. Although these works all explore the common theme of "reasoning on images," our approach to achieving this goal is entirely new. Our method is centered on the inherent grounding and reflection capabilities of Multimodal Large Language Models, and we have built a novel core mechanism and learning method to support it.
>
> The main differences can be summarized below:
>
> ### **1.Comparison with ReFocus:**
> The primary distinction of ChartSketcher is that it leverages the end-to-end visual grounding capability of MLLM itself to draw sketches, and it also provides image feedback and reflection capabilities, offering integration and interactivity that are not present in the other two works. Refocus are semi-"training-free" solutions that rely on external tools for grounding and do not possess reflection capabilities. The main differences can be summarized in the following four aspects:
>
> 1.1 **Difference in Idea**: ChartSketcher utilizes its own learned grounding ability to externalize its reasoning process through "addition" (drawing sketches), whereas ReFocus relies on external tools to acquire coordinates and simplifies the input through "subtraction" (masking information).
>
> 1.2 **Difference in Capability**: ChartSketcher's own grounding and reasoning capabilities enable it to "inspect, reflect, and correct" its own drawings, forming a closed-loop for error correction. In contrast, ReFocus's reliance on external tools makes its process linear, focusing on preventing errors, reducing information redundancy, and focusing on content.
>
> 1.3 **Difference in Technical Route**: ChartSketcher builds the model's own end-to-end grounding capability from scratch through a two-stage training process to provide robust reasoning. In contrast, ReFocus uses prompting or fine-tuning to teach the model how to use an external tool to achieve clear reasoning.
>
> 1.4 **Difference in Implementation Cost**: ChartSketcher constructs its reasoning data using locally deployed open-source models, while ReFocus uses proprietary APIs to build its data.
>
> ### **2.Comparison with Visual Sketchpad:**
>
> Overall, ChartSketcher internalizes visual reasoning and self-correction capabilities as the model's own core skills through a specialized training pipeline, whereas Visual Sketchpad positions the model as a zero-shot planner whose capability is demonstrated in the effective orchestration of external specialist tools.
>
> 2.1 **Difference in Idea:** ChartSketcher uses internal cognitive emulation - teaching models to generate primitive drawing commands for intrinsic visual reasoning and self-correction. Visual Sketchpad uses external tool orchestration - using MLLMs as planners to invoke specialist models.
>
> 2.2 **Difference in Capability:** ChartSketcher's core capability is self-correction through training to inspect visual outputs and generate corrective actions. Visual Sketchpad excels at task delegation, decomposing problems and assigning them to appropriate external tools.
>
> 2.3 **Difference in Technical Route:** ChartSketcher employs a novel two-stage training method with cross-modal distillation and RL. Visual Sketchpad uses zero-shot prompting without fine-tuning, leveraging pre-existing tool-use abilities.
>
> ## question2:
> ## reply:
> Thank you for your suggestions; it will help us improve the transparency of our dataset. In line 74, we stated that 300k data points were used for cold start and 50k for RL, and in line 219, we mentioned using 20% VisualCoT data, but this was not precise enough. We will add a new section to the Appendix detailing the sources of our training data. Specifically, we used a total of 325,668 data points during the cold start phase and 50,455 data points during the RL phase. The specific breakdown of the datasets is fully listed below.
>
> | Training Phase | Method | Data Source                 | Data Type                 | Quantity         |
> | :-- | :----- | :----- | :----- | :- |
> | Cold Start     | SFT    | EvoChart Synthetic Chart Data | Correct Reasoning Path    | 155203 (87.3%)   |
> |                |        | VisualCoT and its Annotations | Correct Reasoning Path    | 22510 (12.7%)    |
> |                |        | Total                       |                           | 177713           |
> |                | DPO    | EvoChart Synthetic Chart Data | Reflection Reasoning Path | 147955           |
> | RL             | KTO    | ChartQA and ChartBench      | MCTS Sampled Paths        | 41196 (81.6%)    |
> |                |        | General QA-Pairs \*          | MCTS Sampled Paths        | 9259 (18.4%)     |
> |                |        | Total                       |                           | 50455            |
>
> \* 18.4% of the KTO training data was derived from general vision-language QA-pairs. These were sourced from datasets aggregated by VisualCoT (*TextVQA, TextCaps, DocVQA, DUDE, SROIE, CUB-200-2011, Flickr30k, Visual7W, InfographicsVQA, VSR, GQA, and OpenImages*). For these samples, we only used their image and QA-pair without adopting the original annotations from VisualCoT, which is effectively equivalent to using the datasets listed above. In the main text, this collection was abbreviated as 'VisualCoT' to save space, and we provide individual citations for each of these datasets in the appendix.
>
> ## weakness 3,4 & question 3,4:
> Thank you again for your in-depth analysis of our work and your valuable suggestions for revision.
>
> While the high-quality dataset we built is a valuable outcome of our research, the central scientific claim of our paper—as outlined in Lines 71-79—is exclusively about the **“ChartSketcher method”**. The effectiveness of each component of ChartSketcher has been proven through ablation experiments. Our experiments involving general VQA data were designed with a specific purpose: to test the hypothesis that this method generalizes across domains, **not to suggest** that the “chart dataset” itself “helps general VQA tasks”. We will revise the manuscript to make this distinction even more explicit.
>
> Nevertheless, you raise a valuable point, as it helps to precisely clarify the role and contribution of our dataset. To strongly address your concerns and to more precisely quantify the generalization ability of our method, we have strictly followed your recommendations and completed these critical ablation and baseline experiments on a small-scale ChartSketcher-2B model.
>
> | method            | ChartQA-H | ChartQA-A | visCoT  |
> |-|-|-|-|
> | Qwen2vl 2B        | 50.48     | 75.84     | 58.33   |
> | ChartSketcher-2B  | 55.60     | 80.88     | 66.86   |
> | w/o VisCoT        | 56.72     | 81.76     | 56.10   |
> | w/o Chart Data    | 49.60     | 74.48     | 64.81   |
>
> The results of these experiments clearly reveal the following points:
>
> 1.	When the model is trained only with our chart data, it still maintains strong performance on chart tasks. This demonstrates the effectiveness of our method and data for chart tasks.
>
> 2.	When the model is trained only with Visual-CoT data, due to the small amount of data, we observed that the Chart capability did not decrease significantly compared to Qwen2vl, and the VisualCoT capability showed a slight improvement.
>
> 3.	Our full model outperforms the aforementioned baseline model in general tasks. This strongly proves that the ChartSketcher method can be used for natural images. And it proves that there might be potential synergistic effects between data from different domains.
>
> In Appendix F, Figure 9, we also demonstrate the intuitive results of applying the ChartSketcher method to natural images.

---

> > ### Comment · Reviewer_hekG · 2025-08-04
> >
> > Thanks for authors' response. My questions are addressed. I have increased my score.

---

> > > ### Author Response · Authors · 2025-08-05
> > >
> > > We sincerely appreciate you taking the time to review our rebuttal and for increasing your score. Your recognition of our work is a great encouragement to us.

---

> ### Comment · Area_Chair_ZQXP · 2025-08-03
>
> Dear Reviewer,
>
> Could you please check if the authors’ rebuttal adequately addresses your concerns? If so, kindly acknowledge the rebuttal and provide any additional comments. If not, it would be greatly appreciated it if you could engage in a discussion with the authors. Your input at this stage is essential to the review process. Thank you very much for your time and effort!
>
> AC

---

### Official Review · Reviewer_QsY7 · 2025-07-03

**Clarity:** 3
**Significance:** 3
**Originality:** 3
**Rating:** 4
**Confidence:** 4

**Summary:**

This paper introduces ChartSketcher, a method for improving multimodal large language models (MLLMs) in chart understanding tasks. The core idea is to enable step-by-step visual reasoning by equipping MLLMs with a programmatic sketching interface (Sketch-CoT), allowing them to annotate, reflect, and correct their reasoning on chart images through iterative multimodal feedback. ChartSketcher employs a two-stage training scheme: first, a 'cold start' phase with cross-modal distillation and stepwise reasoning pattern acquisition from synthetic annotated data, followed by off-policy reinforcement learning (RL) with a Monte Carlo Tree Search (MCTS) variant, to refine reflection and error correction capabilities. Empirical evaluations show that ChartSketcher improves performance over competitive baselines on a range of chart-specific and general vision-language benchmarks, and ablation studies clarify the importance of its feedback, reflection, and RL components.

**Questions:**

See the weaknesses

**Ethical Concerns:**

["NO or VERY MINOR ethics concerns only"]

**Final Justification:**

After careful review of the author's revisions and responses, I confirm that **all concerns raised in my initial assessment have been adequately addressed**. The rebuttal significantly strengthened the manuscript's methodology, clarified key ambiguities, and improved the overall coherence of the argument. I recommend this paper for acceptance.

**Quality:**

3

**Strengths And Weaknesses:**

*Strengths*

1. ChartSketcher presents a mechanism for visual sketching during multimodal CoT reasoning, providing a clear methodological advance over text-only or region-cropping based approaches. The method grounds abstract reasoning into explicit visual steps, as well-illustrated in Figure 1, which shows the circular process of solution, reflection, correction, and continued multimodal feedback.

2. Extensive benchmarks are reported in Table 1, demonstrating tangible improvements versus strong open-source and proprietary models (e.g., Qwen2VL-72B, QvQ-Preview, and even GPT-4o) on chart-specific datasets, with good transfer to general vision-language QA tasks. The inclusion of ablation studies provides further insight into the importance of each architectural and training component.

3. This work describes the creation of a substantial annotated multimodal reasoning dataset, which is a significant contribution for future research in this area.

*Weaknesses*

1. While the empirical advances are clear, the paper lacks a deep theoretical analysis or guarantees about the convergence/robustness of the multimodal sketch-feedback loop or the off-policy RL setup beyond empirical success. Readers might expect some formal theoretical justification or deeper analysis of why Sketch-CoT and visual feedback provide improvements, versus being primarily empirically driven innovations.

2. The feasibility of scaling the programmatic sketching framework and feedback pipeline to more complex, real-world chart styles (e.g., 3D charts, interactive/time-varying charts, annotated infographics) is not systematically explored. More benchmarks can be evaluated, such as ChartX and CharXiv.

3. More baselines can be supplemented to increase the impact in the field of Chart Understanding, e.g., StructChart,ChartVLM, ChartAst, OneChart

---

> ### Author Rebuttal · Authors · 2025-07-31
>
> ## overall:
> Thank you very much for your detailed review of our paper, and for your valuable feedback. We are pleased that you recognized the novelty of our method, our extensive experiments, and the value of our contribution of a high-quality annotated dataset to the community.
>
> We have carefully considered each of the weaknesses you pointed out. In our response, we will address your concerns regarding theoretical depth, benchmark diversity, and the inclusion of additional baseline models one by one.
>
> ## weakness 1:
> ## reply:
> Thank you for your valuable suggestion. Our work is situated within a significant method that draws inspiration from complex human cognitive processes to address current challenges in chart understanding.
>
> Indeed, in the domain of cognitively-inspired AI, many pioneering works establish their contributions primarily through robust empirical validation. This is often because a formal theoretical analysis of such complex, interactive systems—especially those involving Large Language Models—presents a formidable open challenge. Prominent recent examples, such as "Mind's Eye of LLMs", "Enhancing LLM Reasoning via Vision-Augmented Prompting", and "Goal Reduction with Loop-Removal...", all exemplify this approach. They introduce novel frameworks inspired by human cognition and demonstrate their efficacy and potential through extensive empirical success, which constitutes their primary contribution.
>
> While a formal theoretical analysis is exceptionally challenging for such complex, cognitively-inspired systems, our comprehensive experiments robustly demonstrate the effectiveness of the proposed ChartSketcher method. We believe that establishing this empirical success is a crucial and indispensable first step that validates the potential of this paradigm, paving the way for future theoretical investigation. In our revised manuscript, we will explicitly state this as a limitation in the conclusion and highlight it as a key avenue for future research.
>
> [1]Mind's Eye of LLMs: Visualization-of-Thought Elicits Spatial Reasoning in Large Language Models [NeurIPS 2024]
>
> [2]Enhancing LLM Reasoning via Vision-Augmented Prompting [NeurIPS 2024]
>
> [3]Goal Reduction with Loop-Removal Accelerates RL and Models Human Brain Activity in Goal-Directed Learning [NeurIPS 2024]
>
> ## weakness 2:
> ## reply:
> This is a great suggestion. We agree that validating the generalization capability of our framework on more diverse and complex charts is crucial. To address this and further test the robustness of our method, we commit to adding citation and experimental results on the ChartX dataset in the revised version of our paper. We will place the supplementary experimental results table at the end of this paragraph.
>
> | Model | SE (strict) | SE (slight) | SE (high) | QA |
> | :--- | :--- | :--- | :--- | :--- |
> | **ChartSketcher** | 35.65 | 41.74 | 47.06 | 56.94 |
> | **ChartVLM** | 23.18 | 30.68 | 38.30 | 43.84 |
>
> Regarding more advanced types such as 3D or interactive charts, we believe this would require dedicated extensions to our programmatic sketching library and represents a valuable direction for future research. We will discuss this in depth in the Appendix of our paper.
>
> ## weakness3:
> ## reply:
> Thank you for your valuable suggestion. We will cite and discuss in detail the important chart understanding baselines you mentioned in both the "Related Work" section and the appendix of our revised paper.
> Regarding a direct performance comparison, we would like to clarify one point: since the aforementioned models are chart expert models, we will discuss and compare their performance specifically on chart-related tasks in the appendix. We will not evaluate their general-purpose VQA capabilities, as this would constitute an unfair comparison for these baselines, which were not designed for such tasks.
>
> To provide an intuitive reference directly in this response, here is a comparison of ChartSketcher with these works on a representative chart dataset:
>
> | Model | ChartQA (aug) | ChartQA (hum) |
> | :--- | :--- | :--- |
> | **ChartSketcher** | **92.64** | **85.20** |
> | OneChart | 85.30 | 49.10 |
> | ChartAst | 93.90 | 65.90 |
>
> A complete discussion and detailed results on more datasets will be added to the appendix of the paper. We believe this approach satisfies the need for rigor while providing readers with the clearest and fairest performance reference.

---

> > ### Comment · Reviewer_QsY7 · 2025-08-03
> >
> > Thanks for your rebuttal and additional experiments. Since all my concerns have been addressed, I will maintain my positive rating. I also highly recommend the authors cite the references I mentioned before to make the paper in Chart Understanding (CU) field more complete.

---

> > > ### Author Response · Authors · 2025-08-05
> > >
> > > Thank you for your positive feedback and for maintaining your rating, we will include the recommended citations to make our work more complete.

---

> ### Comment · Area_Chair_ZQXP · 2025-08-03
>
> Dear Reviewer,
>
> Could you please check if the authors’ rebuttal adequately addresses your concerns? If so, kindly acknowledge the rebuttal and provide any additional comments. If not, it would be greatly appreciated it if you could engage in a discussion with the authors. Your input at this stage is essential to the review process. Thank you very much for your time and effort!
>
> AC.

---

### Note · Authors · 2025-08-13

Dear Area Chairs and all reviewers,

We sincerely thank you for your invaluable time and insightful feedback during the rebuttal and discussion phases. Our main contribution is ChartSketcher, a novel method where a model reasons by sketching its intermediate steps directly onto an image. This establishes a visual feedback loop, enabling iterative reflection and self-correction.

To directly address the reviewers' concerns, we have conducted extensive additional experiments and are committed to integrating all promised revisions into our final manuscript. These revisions include: (1) New ablation studies to precisely quantify the unique contributions of our dataset and the Sketch-MCTS algorithm; (2) Performance comparisons with more related works and evaluations on more diverse benchmarks (ChartX and more) to better situate our work's value; and (3) More detailed explanations of our methodology, covering the visual feedback loop, data composition, and inference efficiency. We believe these revisions will enhance the paper's rigor and more clearly demonstrate the value and robustness of the ChartSketcher method.

Thank you.

---

### Decision · Program_Chairs · 2025-09-17

**Decision:**

Accept (poster)

**Comment:**

This paper introduces a method to encourage multimodal large language models to perform feedback-driven, step-by-step reasoning. The main idea is a programmatic sketching interface (Sketch-CoT), which allows models to annotate intermediate reasoning steps directly onto charts using a programmatic sketching library. The authors also design a two-stage training procedure: a cold-start phase and RL tuning. Experimental results show promising improvements.

Some reviewers raised concerns about novelty and requested further comparison to existing work, specifically “ReFocus: Visual Editing as a Chain of Thought for Structured Image Understanding” (ICML 2025). The authors provided additional results and explanations during the rebuttal, and the reviewers seemed satisfied. There were also minor concerns about implementation details and conceptual clarifications, which the authors addressed adequately.

Overall, there are no critical drawbacks to this paper. However, while the reviewers appreciated the contributions, they did not find them particularly impressive. The AC considers this a borderline paper, leaning slightly toward acceptance.